# Accelerated Learning with Linear Temporal Logic using Differentiable Simulation

**Alper Kamil Bozkurt**[*]
Electrical & Computer Engineering
Virginia Commonwealth University
Richmond, VA, USA
`bozkurta@vcu.edu`

**Calin Belta**[†] **& Ming C. Lin**[†]
Department of Computer Science
University of Maryland
College Park, MD, USA
`{calin,lin}@umd.edu`

## Abstract

Ensuring that reinforcement learning (RL) controllers satisfy safety and reliability constraints in real-world settings remains challenging: state-avoidance and constrained Markov decision processes often fail to capture trajectory-level requirements or induce overly conservative behavior. Formal specification languages such as linear temporal logic (LTL) offer correct-by-construction objectives, yet their rewards are typically sparse, and heuristic shaping can undermine correctness. We introduce, to our knowledge, the first end-to-end framework that integrates LTL with differentiable simulators, enabling efficient gradient-based learning directly from formal specifications. Our method relaxes discrete automaton transitions via soft labeling of states, yielding differentiable rewards and state representations that mitigate the sparsity issue intrinsic to LTL while preserving objective soundness. We provide theoretical guarantees connecting Büchi acceptance to both discrete and differentiable LTL returns and derive a tunable bound on their discrepancy in deterministic and stochastic settings. Empirically, across complex, nonlinear, contact-rich continuous-control tasks, our approach substantially accelerates training and achieves up to twice the returns of discrete baselines. We further demonstrate compatibility with reward machines, thereby covering co-safe LTL and LTL$_f$ without modification. By rendering automaton-based rewards differentiable, our work bridges formal methods and deep RL, enabling safe, specification-driven learning in continuous domains.

## 1 Introduction

The growing demand for artificial intelligence (AI) systems to operate in a wide range of environments underscores the need for systems that can learn through interaction with their environments, without relying on human intervention. Reinforcement learning (RL) has emerged as a powerful tool for training controllers to perform effectively in uncertain settings with intricate, high-dimensional, and nonlinear dynamics. Despite the promising results in controlled environments, deploying learned controllers in real-world systems–where malfunctioning can be costly or hazardous–requires not only high performance but also strict compliance with formally specified safety and reliability requirements. Therefore, ensuring that learned controllers meet these critical specifications is essential to fully realize the potential of AI systems in real-world applications. Safety in learning is often modeled with constrained Markov decision processes (MDPs) (e.g. Ding et al. (2021)), where the accumulated cost must be within a budget. However, additive cost functions may not reflect real-world safety, as assigning meaningful costs to harms is challenging. Alternative approaches define safety by avoiding unsafe states or actions (e.g. Qin et al. (2021)), which is simpler than designing cost functions. However, this may result in overly conservative policies and could not capture complex trajectory-level requirements.

Recently, researchers have explored specifying RL objectives using formal languages, which explicitly and unambiguously express trajectory-based task requirements, including safety and liveness properties. Among these, linear temporal logic (LTL) has gained particular popularity (e.g. Hahn et al. (2019); Bozkurt et al. (2020b); Icarte et al. (2022); Hasanbeig et al. (2023)) due to the automaton-based memory it offers, which ensures history-independence and makes it especially

---

[*]work done while at University of Maryland, College Park
[†]equal supervising

suitable for long-horizon tasks, unlike other languages such as signal temporal logic (STL). Specifying desired properties in LTL inherently prevents mismatches between the intended behavior and the behavior learned through reward maximization–one of the most well-known safety challenges in AI (Amodei et al., 2016). Although these methods are proven to define the correct RL objectives, the sparse logical rewards make learning extremely difficult, as obtaining a nonzero reward often requires significant exploration. Denser LTL-based rewards provided through heuristics might accelerate learning (Kantaros, 2022); however, if not carefully designed, they can compromise the correctness of the objective and misguide exploration depending on the environment, ultimately reducing learning efficiency. In this work, we address the challenge of scalable learning with correct objectives for temporally extended tasks. We adopt LTL as the specification language, leveraging the intuitive high-level language and the automaton-based memory it provides. Unlike prior methods, our approach *harnesses gradients from differentiable simulators to facilitate efficient learning directly from LTL specifications*, while preserving the correctness of the objectives. Our contributions can be summarized as follows:

- We propose, to the best of our knowledge, the first approach that accelerates learning from LTL specifications using differentiable simulators. Our approach effectively mitigates the inherent issue of the sparse rewards without sacrificing the expressiveness and correctness that LTL provides.
- We introduce soft labeling techniques for continuous environments that yield probabilistic $\varepsilon$-actions and transitions in the automata derived from LTL, which ensures the differentiability of rewards and states with respect to actions. We establish formal guarantees connecting automata acceptance conditions with our differentiable framework, yielding a tunable bound on the discrepancy between discrete and differentiable LTL rewards, including in stochastic settings.
- We demonstrate that our method accelerates learning, achieving **up to twice the returns** of baselines across diverse experiments in complex, nonlinear, contact-rich settings where standard approaches struggle to learn without handcrafted reward shaping. We further evaluate on reward machines, showing that our differentiable approach generalizes across formal method frameworks.

## 2 RELATED WORK

**Safe RL.** One common perspective in Safe RL defines safety as the guarantee on the cumulative costs over time within a specified safety budget, which is often modeled using constrained MDPs and has been widely studied (García & Fernández, 2015; Chow et al., 2018; Stooke et al., 2020; Ding et al., 2021), relying on additive cost functions and budgets, which may not adequately capture safety in many scenarios. In practice, it is often difficult to assign unambiguous scalar costs reflecting trade-offs between different harmful situations (Skalse et al., 2022). Other approaches define safety in terms of avoiding unsafe states and focus on preventing or modifying unsafe actions via shielding or barrier functions (Berkenkamp et al., 2017; Cheng et al., 2019; Qin et al., 2021; Zhan et al., 2024), requiring only the identification of unsafe states and actions, which is often easier than designing cost functions (Wang et al., 2023); however, they can lead to overly conservative control policies (Yu et al., 2022). Moreover, the requirements are often placed over trajectories, which could be more complex than simply avoiding certain states (Hsu et al., 2021). Our approach avoids these issues by employing LTL as the specification language to obtain correct-by-construction RL objectives.

**RL with Temporal Logics.** There has been increasing interest in using formal specification languages to encode task objectives that are trajectory-dependent, particularly those involving safety requirements. LTL has emerged as a widely adopted formalism due to its expressiveness and well-defined semantics over infinite traces. Initial attempts to combine LTL with RL relied on model-based approaches (Fu & Topcu, 2014a; Wen & Topcu, 2021), which reduce specification satisfaction into a reachability problem that can be solved by RL, by exploiting the MDP transition structure to construct a product MDP with automata derived from LTL. However, the unavailability of accurate transition models limits their applicability, especially in deep RL contexts. Thus, model-free RL methods for LTL emerged, notably reward machines (RMs) (Toro Icarte et al., 2018; Icarte et al., 2018; Camacho et al., 2019; Icarte et al., 2022) for co-safe LTL and $LTL_f$ fragments, which directly generate rewards based on the acceptance states of the derived automata without explicit knowledge of transition dynamics. The introduction of LDBAs for MDP model checking (Hahn et al., 2015), facilitated structured reward design with their simpler acceptance conditions for general LTL formulas (Hahn et al., 2019; Bozkurt et al., 2020b). This line of work inspired numerous extensions and applications across broader domains (Voloshin et al., 2022; 2023; Le et al., 2024; Perez et al., 2024; Yalcinkaya et al., 2024; Jackermeier & Abate, 2025). Researchers have also explored continuous-time logics such as STL, whose robustness scores can be used as rewards (Aksaray et al., 2016).

However, these scores typically depend on historical information, violating the Markov assumption and thereby restricting their use in long-horizon, stochastic, or value-based RL settings. For detailed explanations and comparisons, see Appx. A.

**RL with Differentiable Simulators.** Differentiable simulators enable gradient-based policy optimization in RL by computing gradients of states and rewards with respect to actions, using analytic methods (Carpentier & Mansard, 2018; Geilinger et al., 2020; Qiao et al., 2021; Xu et al., 2021; Werling et al., 2021) or auto-differentiation (Heiden et al., 2021; Freeman et al., 2021). While Back-propagation Through Time (BPTT) is commonly used (Zamora et al., 2021; Du et al., 2021; Huang et al., 2021; Hu et al., 2020; Liang et al., 2019; Hu et al., 2019), it suffers from vanishing or exploding gradients for long horizons as it ignores the Markov property of states (Metz et al., 2021). To address this, several differentiable RL algorithms have been proposed (Parmas et al., 2018; Suh et al., 2022). Short Horizon Actor-Critic (SHAC) (Xu et al., 2022) divides long trajectories into shorter segments where BPTT is tractable and bootstraps the remaining trajectory using the value function. Adaptive Horizon Actor-Critic (AHAC) (Georgiev et al., 2024) extends SHAC by dynamically adjusting the segment lengths based on contact information from the simulator. Gradient-Informed PPO (Son et al., 2023) integrates gradient information to the RL framework in an adaptive manner. Our approach builds a differentiable, Markovian transition function for LTL-derived automata, making it compatible with all differentiable RL methods. Unlike prior STL-based efforts (Leung et al., 2023; Meng & Fan, 2023), which rely on non-Markovian rewards and BPTT, our method supports efficient long-horizon learning with full differentiability.

## 3 PRELIMINARIES AND PROBLEM FORMULATION

**MDPs.** We formalize the interaction between controllers with the environments as MDPs.

**Definition 1.** *A (differentiable) MDP is a tuple $M = (S, A, f, p_0)$ such that $S$ is a set of continuous states; $A$ is a set of continuous actions; $f : S \times A \mapsto S$ is a differentiable transition function; $p_0$ is an initial state distribution where $p_0(s)$ denotes the probability density for the state $s$.*

For instance, for a given robotic task, the state space $S$ is the positions $\mathtt{x}$ and velocities $\dot{\mathtt{x}}$ of relevant objects, body parts, and joints. The action space $A$ may consist of torques applied to the joints. The transition function $f$ captures the underlying system dynamics and outputs the next state via computing the accelerations $\ddot{\mathtt{x}}$ by solving $\mathtt{M}\ddot{\mathtt{x}} = \mathtt{J}^\mathsf{T}\mathtt{F}(\mathtt{x}, \dot{\mathtt{x}}) + \mathtt{C}(\mathtt{x}, \dot{\mathtt{x}}) + \mathtt{T}(\mathtt{x}, \dot{\mathtt{x}}, a)$, for a given state $s = \langle \mathtt{x}, \dot{\mathtt{x}} \rangle \in S$ and action $a \in A$. Here, $\mathtt{M}$ is a mass matrix; and $\mathtt{F}$, $\mathtt{C}$, and $\mathtt{T}$ are, respectively, force, Coriolis, and torque functions that can be approximated using differentiable physics simulators.

**RL Objective.** In RL, a policy $\pi{:}S^+{\mapsto}A$ is evaluated based on the expected cumulative reward, i.e. return, associated with the paths $\sigma{:=}s_0 s_1 \ldots$ (sequence of visited states) generated by the Markov chain (MC) $M_\pi$ induced by the policy $\pi$. We write $\sigma[t]$, $\sigma[:t]$, $\sigma[t:]$ for $s_t$, the prefix $s_0 \ldots s_t$ and the suffix $s_t s_{t+1} \ldots$ For a given reward function $R{:}S^+{\mapsto}\mathbb{R}$, a discount factor $\gamma{\in}(0,1)$ and a horizon $H$, the return of a path $\sigma$ from time $t{\in}\mathbb{N}$, is defined as $G_{t:H}(\sigma){=}\sum_{i=t}^{H} \gamma^i R(\sigma[:i])$. For simplicity, we denote the infinite-horizon return starting from $t{=}0$ as $G_H(\sigma){:=}G_{0:H}(\sigma)$, and further drop the subscript to write $G(\sigma){:=}\lim_{H\to\infty} G_H(\sigma)$. We note that for Markovian reward functions ($R{:}S{\mapsto}\mathbb{R}$), memoryless policies ($\pi{:}S{\mapsto}A$) suffice. However, the tasks we consider require finite-memory policies. To address this, we reduce the problem of obtaining a finite-memory policy to that of learning a memoryless policy by augmenting the state space $S$ with memory states, as detailed in Sec. 4. The discount factor reduces the value of future rewards to prioritize immediate ones: a reward received after $t$ steps contributes $\gamma^t R(\sigma[t])$ to the return. The objective in RL, specifically in policy gradient, is to learn optimal policy parameters $\theta^*{=}\arg\max_\theta J(\theta)$ where $J(\theta){=}\mathbb{E}_{\sigma\sim M_{\pi_\theta}}[G_H(\sigma)]$. In differentiable MDPs, RL can leverage first-order gradients $\nabla_\theta^{[1]} J(\theta){=}\mathbb{E}_{\sigma\sim M_{\pi_\theta}}[\nabla_\theta G_H(\sigma)]$ where $\frac{\partial G_H}{\partial s_t}{=}\frac{\partial G_H}{\partial s_{t+1}}\frac{\partial f}{\partial s_t}$, $\frac{\partial G_H}{\partial a_t}{=}\frac{\partial G_H}{\partial s_{t+1}}\frac{\partial f}{\partial a_t}$ via BPTT (see also Appx. B, C).

**Labels.** We define the set of atomic propositions (APs), denoted by $\mathtt{A}$, as properties of interest that place bounds on functions of the state space. Formally, each AP takes the form $\mathtt{a}{:=}\text{`}g(s){>}0\text{'}$, where $g : S \mapsto \mathbb{R}$ is assumed to be a differentiable function mapping a given state to a signal. For example, the function $g(\langle \mathtt{x}, \dot{\mathtt{x}} \rangle) := \dot{\mathtt{x}}_{\max}^2 - \dot{\mathtt{x}}_i^2$ can be used to define an AP that specifies that the velocity of the $i$-th component must be below an upper bound $\dot{\mathtt{x}}_{\max}$. The labeling function $L : S \mapsto 2^{\mathtt{A}}$ returns the set of APs that hold true for a given state. Specifically, an AP $\mathtt{a} := \text{`}g(s) > 0\text{'}$ is included in the label set $L(s)$ of state $s$ – i.e., $s$ is labeled by $\mathtt{a}$ if and only if (iff) $g(s) > 0$. We also write, with a slight abuse of notation, $L(\sigma) := L(\sigma[0])L(\sigma[1]) \ldots$ to denote the trace (sequences of labels) of a path $\sigma$. Finally, we write $M^+{=}(M, L)$ to denote a labeled MDP.

**LTL.** LTL provides a high-level formal language for specifying the desired temporal behaviors. Alongside the standard operators in propositional logic – negation ($\neg$) and conjunction ($\wedge$) – LTL offers two temporal operators, namely next ($\bigcirc$) and until ($\mathsf{U}$). The formal syntax of LTL is defined by the following grammar (Baier & Katoen, 2008): $\varphi := \text{true} \mid \mathsf{a} \mid \neg\varphi \mid \varphi_1 \wedge \varphi_2 \mid \bigcirc\varphi \mid \varphi_1 \mathsf{U} \varphi_2$, $\mathsf{a} \in A$. The semantics of LTL formulas are defined over paths. Specifically, a path $\sigma$ either satisfies $\varphi$, denoted by $\sigma \models \varphi$, or not ($\sigma \not\models \varphi$). The satisfaction relation is defined recursively as follows: $\sigma \models \varphi$; if $\varphi = \mathsf{a}$ and $\mathsf{a} \in L(\sigma[0])$ (i.e., $\mathsf{a}$ immediately holds); if $\varphi = \neg\varphi'$ and $\sigma \not\models \varphi'$; if $\varphi = \varphi_1 \wedge \varphi_2$ and $(\sigma \models \varphi_1) \wedge (\sigma \models \varphi_2)$; if $\varphi = \varphi_1 \mathsf{U} \varphi_2$ and there exists $t \geq 0$ such that $\sigma[t{:}] \models \varphi_2$ and for all $0 \leq i < t$, $\sigma[i{:}] \models \varphi_1$. The remaining Boolean and temporal operators can be derived via the standard equivalences such as eventually ($\Diamond\varphi := \text{true} \mathsf{U} \varphi$) and always ($\Box\varphi := \neg(\Diamond\neg\varphi)$).

**LDBAs.** If a path satisfies a given LTL formula $\varphi$ can be checked by building an LDBA, denoted by $\mathcal{A}_\varphi$ that is suitable for quantitative model-checking of MDPs (Sickert et al., 2016). An LDBA is a tuple $\mathcal{A}_\varphi = (Q, q_0, \Sigma, \delta, B)$ where $Q$ is a finite set of states; $q_0 \in Q$ is the initial state; $\Sigma = 2^A$ is the set of labels; $\delta : Q \times (\Sigma \cup \{\varepsilon\}) \mapsto 2^Q$ is a transition function triggered by labels; $B \subseteq Q$ is the accepting states. An LDBA $\mathcal{A}_\varphi$ accepts a path $\sigma$ (i.e., $\sigma \models \varphi$), iff its trace $L(\sigma)$ induces an LDBA execution visiting some of the accepting states infinitely often, known as the Büchi condition (see Appx. D).

**LTL Learning Problem.** Our objective is to learn control policies that ensure given path specifications are satisfied by a given labeled MDP. In stochastic environments, this objective translates to maximizing the probability of satisfying those specifications. We consider specifications given as LTL formulas since LTL provides a high-level formalism well-suited for expressing safety and other temporal constraints in robotic systems–and, importantly, finite-memory policies suffice to satisfy LTL specifications (Chatterjee & Henzinger, 2012). We now formalize the problem as follows:

**Problem 1.** *Given a labeled MDP $M^+$ and a LTL formula $\varphi$, find an optimal finite-memory policy $\pi_\varphi^*$ that maximizes the probability of satisfying $\varphi$, i.e., $\pi_\varphi^* := \underset{\pi \in \Pi}{\arg\max} \Pr_{\sigma \sim M_\pi^+} \{\sigma \mid \sigma \models \varphi\}$, where $\Pi$ is the set of policies and $\sigma$ is a path drawn from the Markov chain (MC) $M_\pi^+$ induced by $\pi$.*

## 4 ACCELERATED LEARNING FROM LTL USING DIFFERENTIABLE REWARDS

In this section, we present our approach for efficiently learning optimal policies that satisfy given LTL specifications by leveraging differentiable simulators. We first define product MDPs and discuss their conventional use in generating discrete LTL-based rewards for RL. We then introduce our method for deriving differentiable rewards using soft labeling, enabling gradient-based optimization while preserving the logical structure of the specifications. We lastly establish a theorem yielding a tunable bound on the discrepancy between discrete and differentiable LTL rewards.

**Product MDPs.** A product MDP is constructed by augmenting the states and actions of the original MDP with indicator vectors representing the LDBA states. The state augmentations serve as memory modes necessary for tracking temporal progress, while the action augmentations, referred to as $\varepsilon$-actions, capture the nondeterministic $\varepsilon$-moves of the LDBA. The transition function of the product MDP reflects a synchronous execution of the LDBA and the MDP; i.e., upon taking an action, the MDP moves to a new state according to its transition probabilities, and the LDBA transitions by consuming the label of the current MDP state.

**Definition 2.** *A product MDP is a tuple $\mathbf{M} = (\mathbf{S}, \mathbf{A}, \mathbf{f}, \mathbf{p_0}, \mathbf{B})$, composed of a labeled MDP $M^+ = (S, A, f, p_0, \mathsf{A}, L)$ and an LDBA $\mathcal{A}_\varphi = (Q, \Sigma = 2^\mathsf{A}, \delta, q_0, B)$ derived from an LTL formula $\varphi$ such that $\mathbf{S} = S \times \mathbf{Q}$ is the set of product states and $\mathbf{A} = A \times \mathbf{Q}$ is the set of product actions where $\mathbf{Q} = [0, 1]^{|Q|}$ is the space set for the one-hot indicator vectors of automaton states; $\mathbf{f}: \mathbf{S} \times \mathbf{A} \mapsto \mathbf{S}$ is the transition function defined as*

$$\mathbf{f}(\langle s, \mathbf{q}^q \rangle, \langle a, \mathbf{q}^{q_\varepsilon} \rangle) := \begin{cases} \langle s', \mathbf{q}^{q'} \rangle & q_\varepsilon \notin \delta(q', \varepsilon) \\ \langle s', \mathbf{q}^{q_\varepsilon} \rangle & q_\varepsilon \in \delta(q', \varepsilon) \end{cases} \tag{1}$$

*for given $s, s' \in S$, $a \in A$ and the indicator vectors $\mathbf{q}^q, \mathbf{q}^{q'}, \mathbf{q}^{q_\varepsilon} \in \mathbf{Q}$ for $q, q', q_\varepsilon \in Q$, respectively, where $s' := f(s, a)$ and $q' := \delta(q, L(s))$; $\mathbf{p_0}$ is the initial product state distribution where $p_0^\times(\langle s, \mathbf{q}^q \rangle)[q = q_0]$; $\mathbf{B} = \{\langle s, \mathbf{q}^q \rangle \in \mathbf{S} \mid q \in B\}$ is the set of accepting product states. A product MDP is said to accept a product path $\boldsymbol{\sigma}$ iff $\boldsymbol{\sigma}$ satisfies the Büchi condition, denoted as $\boldsymbol{\sigma} \models \Box\Diamond\mathbf{B}$, which is to visit some states in $\mathbf{B}$ infinitely often.*

By definition, any product path accepted by the product MDP corresponds to a path in the original MDP that satisfies the acceptance condition of the LDBA. Consequently, the satisfaction of the LTL specification $\varphi$ is reduced to ensuring acceptance in the product MDP. This reduces Problem 1 to maximizing the probability of reaching accepting states infinitely often in the product MDP:

**Lemma 1** (from Theorem 3 in (Sickert et al., 2016)). *A memoryless product policy $\boldsymbol{\pi}_\varphi^*$ that maximizes the probability of satisfying the Büchi condition in a product MDP $\mathbf{M}$ constructed from a given labeled MDP $M^+$ and the LDBA $\mathcal{A}_\varphi$ derived from a given LTL specification $\varphi$, induces a policy $\pi_\varphi^*$ with a finite-memory captured by $\mathcal{A}_\varphi$ maximizing the satisfaction probability of $\varphi$ in $M^+$.*

**Discrete LTL Rewards.** The idea is to derive LTL rewards from the acceptance condition of the product MDP to train control policies via RL approaches. Specifically, we consider the approach proposed in (Bozkurt et al., 2024) that uses carefully crafted rewards and state-dependent discounting based on the Büchi condition such that an optimal policy maximizing the expected return is also an objective policy $\boldsymbol{\pi}_\varphi^*$ maximizing the satisfaction probabilities as defined in Lemma 1, which can be formalized as below:

**Theorem 1.** *For a given product MDP $\mathbf{M}$, the expected return for a policy $\boldsymbol{\pi}$ approaches the probability of satisfying the Büchi acceptance condition as the discount factor $\gamma$ goes to 1; i.e., $\lim_{\gamma \to 1^-} \mathbb{E}_{\boldsymbol{\sigma} \sim \mathbf{M}_{\boldsymbol{\pi}}}[G(\boldsymbol{\sigma})] = Pr_{\boldsymbol{\sigma} \sim \mathbf{M}_{\boldsymbol{\pi}}}(\boldsymbol{\sigma} \models \Box\Diamond\mathbf{B})$; if the return $G(\boldsymbol{\sigma})$ is defined as follows:*

$$G(\boldsymbol{\sigma}) := \sum_{t=0}^{\infty} R(\boldsymbol{\sigma}[t]) \prod_{i=0}^{t-1} \Gamma(\boldsymbol{\sigma}[i]), \quad R(\mathbf{s}) := \begin{cases} 1-\beta & \mathbf{s} \in \mathbf{B} \\ 0 & \mathbf{s} \notin \mathbf{B} \end{cases}, \quad \Gamma(\mathbf{s}) := \begin{cases} \beta & \mathbf{s} \in \mathbf{B} \\ \gamma & \mathbf{s} \notin \mathbf{B} \end{cases} \tag{2}$$

*where $\prod_{i=0}^{-1} := 1$, $\beta$ is a function of $\gamma$ satisfying $\lim_{\gamma \to 1^-} \frac{1-\gamma}{1-\beta} = 0$, $R : \mathbf{S} \mapsto [0,1)$ and $\Gamma : \mathbf{S} \mapsto (0,1)$ are state-dependent reward and the discount functions respectively.*

The proof can be found in (Bozkurt et al., 2024). The idea is to encourage the agent to repeatedly visit an accepting state as many times as possible by assigning a larger reward to the accepting states. Further, the rewards are discounted with a larger factor in non-accepting states to reflect that the number of visitations to non-accepting states are not important. The LTL rewards provided by this approach, however, are very sparse; depending on the environment and the structure of the automaton, the agent might need to blindly explore a large portion of the state space before getting a nonzero reward, which constitutes the main hurdle in learning from LTL specifications.

**Differentiable LTL Rewards.** We propose employing differentiable RL algorithms and simulators to mitigate the sparsity issue and accelerate learning. However, the standard LTL rewards described earlier are not only sparse but discrete, rendering them non-differentiable with respect to states and actions. This lack of differentiability primarily stems from two factors: the binary state-based reward function and discrete automaton transitions. To address this challenge, we introduce probabilistic "soft" labels. We start by defining the probability that a given AP, denoted as $\mathtt{a} := \text{'}g(s) > 0\text{'}$, belongs to the label $L(s)$ of a state $s$. Formally:

$$\Pr(\mathtt{a} \in L(s)) = \Pr(g(s) > 0) := h(g(s)) = \frac{1}{1 + \exp(-g(s))}. \tag{3}$$

Although we use the widely adopted sigmoid function here[1], any differentiable cumulative distribution function (CDF) $h : \mathbb{R} \mapsto [0,1]$ could be applied. Building upon these probabilities, we define the probability associated with a label $l$ as follows:

$$\Pr(L(s) = l) = \prod_{\mathtt{a} \in l} \Pr(\mathtt{a} \in L(s)) \prod_{\mathtt{a} \notin l} (1 - \Pr(\mathtt{a} \in L(s))). \tag{4}$$

These probabilistic labels induce probabilistic automaton transitions, causing the controller to observe automaton states probabilistically. Consequently, instead of modeling automaton states as deterministic indicator vectors in product MDPs, we represent them as probabilistic superpositions over all possible automaton states. By doing so, we design differentiable transitions and rewards within the product MDP. Let $f_L : S \times \mathbf{Q} \mapsto \mathbf{Q}$ denote the function that updates the automaton state probabilities based on the LDBA transitions triggered by probabilistic labels, and let $\mathbf{q}$ denote the vector where each element $\mathbf{q}_q$ is the probability of being in automaton state $q$, then we can define:

$$f_L(\langle s, \mathbf{q} \rangle) = \mathbf{q}' \quad \text{where} \quad \mathbf{q}'_{q'} = \sum_q \mathbf{q}_q \sum_{l \in L_{q,q'}} \Pr(L(s) = l) \quad \text{and} \quad L_{q,q'} := \{l \mid q' = \delta(q,l)\}. \tag{5}$$

Intuitively, the probability of transitioning to a subsequent automaton state $q'$ is computed by summing probabilities across all current automaton states $q$ and labels $l \in L_{q,q'}$ capable of leading to state $q'$. This computation can be efficiently done through differentiable matrix multiplication.

The remaining hurdle is the binary $\varepsilon$-actions available to the controller, which trigger $\varepsilon$-transitions in the LDBA. Similarly to the soft labels approach, $\varepsilon$-actions can become differentiable by representing

---

[1]For the correctness of LTL, $\Pr(g(s) > 0)$ must be exactly 0 or 1 for values below or above certain thresholds. In practice, this is not an issue, as overflow behavior of sigmoid ensures this condition is satisfied.

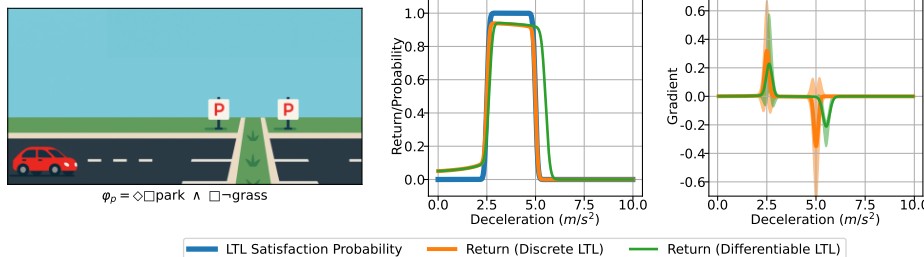

Figure 1: **LTL Returns and Derivatives.** *Left*: The parking scenario where the car must brake to stop in the parking area without entering the grass field ($\varphi_p$). *Middle*: LTL satisfaction probability and return estimates from discrete and differentiable LTL formulations as functions of deceleration. *Right*: LTL return gradients with respect to deceleration and their standard deviation. The key challenge in learning from LTL arises from slightly-sloped regions and sharp changes in the returns produced by discrete LTL rewards. Our *differentiable LTL* approach not only *smooths these abrupt changes but also enables the use of low-variance first-order gradient estimates essential for effective learning in slightly-sloped regions.*

the probabilities of the $\varepsilon$-transitions to be triggered. Let $f_\varepsilon : \mathbf{Q} \times \mathbf{Q} \mapsto \mathbf{Q}$ denote the function updating automaton state probabilities based on the $\varepsilon$-action taken, and let $\mathbf{q}^\varepsilon$ denote the vector whose elements indicate the probabilities of taking the $\varepsilon$-actions leading to the corresponding automaton states, we then define:

$$f_\varepsilon(\mathbf{q}, \mathbf{q}^\varepsilon) = \mathbf{q}' \text{ where } \mathbf{q}'_{q'} = \sum_{q \in Q_{\varepsilon,q'}} \mathbf{q}_q \mathbf{q}^\varepsilon_{q'} + \sum_{q \in \overline{Q}_{q',\varepsilon}} \mathbf{q}_{q'} \mathbf{q}^\varepsilon_q, \ Q_{\varepsilon,q'} := \{q \mid q' \in \delta(q,\varepsilon)\}, \ \overline{Q}_{q',\varepsilon} := \{q \mid q \notin \delta(q',\varepsilon)\}. \quad (6)$$

Conceptually, the probability of transitioning to automaton state $q'$ involves two scenarios: (the first summation in (6)) the probability of moving to $q'$ via valid $\varepsilon$-transitions, and (the second summation in (6)) the probability of remaining in $q'$ after trying to leave from $q'$ via nonexistent $\varepsilon$-transitions. These vector computations can be efficiently performed in a differentiable manner. We can formulate the complete transition function $\mathfrak{f}$ by composing $f_L$, $f_\varepsilon$, and $f$ as follows:

$$\mathfrak{f}(\langle s, \mathbf{q} \rangle, \langle a, \mathbf{q}^\varepsilon \rangle) := \langle f(s,a), f_L(\langle s, f_\varepsilon(\mathbf{q}, \mathbf{q}^\varepsilon) \rangle) \rangle. \quad (7)$$

This transition function first executes the $\varepsilon$-actions, then performs the LDBA transitions triggered by state labels to update the automaton state probabilities, while applying the given action to update the MDP states. The function $\mathfrak{f}$ is fully differentiable with respect to $s$, $\mathbf{q}$, $a$, and $\mathbf{q}^\varepsilon$. We can now obtain a reward $\mathfrak{R} : \mathbf{Q} \mapsto (0,1)$ and a discounting function $\mathfrak{D} : \mathbf{Q} \mapsto (0,1)$ that are also differentiable with respect to states and actions as follows:

$$\mathfrak{R}(\langle s, \mathbf{q} \rangle) := (1-\beta) \sum_{q \in B} \mathbf{q}_q, \quad \mathfrak{D}(\mathbf{q}) := \beta \sum_{q \in B} \mathbf{q}_q + \gamma \sum_{q \notin B} \mathbf{q}_q \quad (8)$$

These differentiable reward and discounting functions allow us to obtain first-order gradient estimates $\nabla^1_\theta J(\theta) := \mathbb{E}_{\sigma \sim M_{\pi_\theta}}[\nabla_\theta G_H(\sigma)]$ which are known to exhibit lower variance compared to zeroth-order estimates (Xu et al., 2022). Such first-order estimates can be effectively utilized by differentiable RL algorithms to accelerate learning. In the following example, we illustrate employing these lower-variance gradient estimates is particularly crucial when learning from LTL rewards.

**Parking Example.** Consider a parking scenario in which the vehicle starts with an initial velocity of $v_0 = 10$ m/s. The controller applies the brakes with a constant deceleration $a \in [0$ m/s$^2, 10$ m/s$^2]$ over the next 10 seconds, with the goal of bringing the car to rest inside the parking area. For safety, the vehicle must not enter the grass field before reaching the parking zone on the right-hand side. We formalize these requirements in LTL as $\varphi_p = \Diamond\Box \texttt{park} \land \Box\neg\texttt{grass}$ where the parking area and the grass field are defined as $\texttt{park} := (x{>}10$ m $\land x{<}20$ m$) \lor (x{>}30$ m $\land x{<}40$ m$)$ and $\texttt{grass} := x{>}20$ m $\land x{<}30$ m, respectively.

Fig. 1 illustrates this task, including satisfaction probabilities, returns, and gradients with respect to deceleration. The satisfaction probability is 1 for deceleration values between 2.5 m/s$^2$ and 5.0 m/s$^2$, and 0 outside this range. The differentiable LTL returns closely match the discrete ones, except near the boundaries of the satisfaction region, where the differentiable version produces smoother transitions. This smoothness is particularly evident in the gradient plots. Although differentiable LTL rewards yield smoother return curves, learning remains challenging due to the small gradient magnitudes across most of the parameter space except near the satisfaction boundaries. For instance, in the region between 0.0 m/s$^2$ and 2.5 m/s$^2$, the returns increase with deceleration, but noisy gradient estimates can still lead the learner away from the satisfaction region. Therefore, obtaining

low-variance gradient estimates is especially beneficial when learning from LTL, where most of the landscape requires sharper gradients for effective optimization. See Appx. E for comparison.

**Discrete vs. Differentiable LTL Rewards.** We now show that the maximum discrepancy between the discrete and differentiable values can be upper-bounded for a given tolerance parameter $\zeta$ and an activation function $h$ in the theorem below. Since, by Theorem 1, the discrete values converge to the satisfaction probabilities, the bound is also valid in the limit for the actual satisfaction probabilities.

**Theorem 2.** *Let $\varsigma$ be the tolerance on the signal bounds of atomic propositions, and let $p$ be the probability associated with $\varsigma$ (i.e., $p := Pr(\varsigma > 0) = h(\varsigma)$) as in (3). Let $G^{disc.}$ and $G^{diff.}$ denote the returns obtained via discrete and differentiable rewards, respectively. Then the maximum discrepancy between them is upper bounded as:*

$$|G^{disc.}(\sigma) - G^{diff.}(\sigma)| < \frac{1}{1 + \frac{1-\beta}{(1-p)^{|A|}}} = \frac{1}{1 + \frac{1-\beta}{(1-h(\varsigma))^{|A|}}} \tag{9}$$

*where $\beta$ is the discount factor for accepting states, as defined in (8). By linearity of expectation, this result immediately extends to the expected return (values) in stochastic environments; i.e., the upper bound above holds for $|\mathbb{E}[G^{disc.}(\sigma)] - \mathbb{E}[G^{diff.}(\sigma)]|$ where expectations are over trajectories drawn under any given policy.*

*Proof.* The maximum discrepancy between $G^{disc.}$ and $G^{diff.}$ occurs when all probabilistic transitions associated with soft labels yield positive differentiable rewards while their corresponding discrete rewards are zero (or vice versa). For a given tolerance $\varsigma$, the probability of incorrectly evaluating all atomic propositions under soft labels is $\rho := (1 - p)^{|A|}$, where A denotes the set of all atomic propositions defined in the LTL grammar. In this worst-case scenario, the differentiable return for a trajectory $\sigma$, where all such transitions lead to accepting states, is:

$$G^{diff.}(\sigma) \leq \sum_{t}^{\infty} \rho(1-\rho)^t \beta^t = \frac{\rho}{1 - (1-\rho)\beta} = \frac{\rho}{(1-\beta) + \rho\beta} = \frac{1}{1 + \frac{1-\beta}{\rho}} = \frac{1}{1 + \frac{1-\beta}{(1-p)^{|A|}}}.$$

Since $G^{disc.} = 0$, this expression provides the upper bound on the maximum discrepancy. $\square$

## 5 EXPERIMENTS

In this section, through simulated experiments, we show learning from differentiable LTL rewards offered by our method is significantly faster than learning from discrete LTL rewards.

**Implementation Details.** We implemented our approach in Python utilizing the PyTorch-based differentiable physics simulator dFlex introduced in Xu et al. (2022). We used an NVIDIA GeForce RTX 2080 GPU, 4 Intel(R) Xeon(R) Gold 5218 CPU cores, and 32 gigabytes memory for each experiment. Specifically, we generate the automaton description using Owl (Kretínský et al., 2018) and parse it using Spot (Duret-Lutz et al., 2016). We then construct reward and transition tensors from the automata. We then compute the probabilities for each observation as explained in the previous section using a sequence of differentiable vector operations via PyTorch. Lastly, using the constructed transition and reward tensors, we update the automaton states and provide rewards. The overall approach is summarized in Algorithm 1.

---

**Algorithm 1** Differentiable RL with LTL

**Require:** MDP $M$, LTL formula $\varphi$, Policy $\pi_\theta$
  Derive LDBA $A_\varphi$ and APs A from $\varphi$
  Derive $\mathfrak{f}$ (7) and $\mathfrak{R}$, $\mathfrak{D}$ (8) from $A_\varphi$
  **while** True **do**
    **for** $i = 1, 2, ..., N$ **do**
      Initialize $\mathbf{q}^{(0)} \sim A_\varphi$, $s^{(0)} \sim M$, $G \leftarrow 0$
      **for** $t = 1, 2, ..., H$ **do**
        Get action $\langle a, \mathbf{q}^\varepsilon \rangle \sim \pi_\theta(\langle s^{(t-1)}, \mathbf{q}^{(t-1)} \rangle)$
        Execute $\varepsilon$-action $\mathbf{q}' \leftarrow f_\varepsilon(\mathbf{q}, \mathbf{q}^\varepsilon)$
        Execute label transition $\mathbf{q}^{(t)} \leftarrow f_L(\langle s, \mathbf{q}' \rangle)$
        Execute MDP action $s^{(t)} \leftarrow f(s, a)$
        Compute reward $r \leftarrow \mathfrak{R}(\mathbf{q}^{(\mathbf{t})})$
        Update return $G_t^{(i)} \leftarrow G_{t-1}^{(i)} + \mathfrak{D}(\mathbf{q}) \cdot r$
      **end for**
    **end for**
    Calculate $\hat{\nabla}_\theta^{[1]} J(\theta) \leftarrow \frac{1}{N} \sum_{i=1}^{N} \nabla_\theta G_H^{(i)}$
    Train $\pi_\theta$ using $\hat{\nabla}_\theta^{[1]} J(\theta)$
  **end while**

---

**Baselines.** We use two widely adopted and representative state-of-the-art (SOTA) model-free RL algorithms as our baseline non-differentiable RL methods ($\partial\!\!\!/$RLs): the on-policy Proximal Policy Optimization (PPO) (Schulman et al., 2017) and the off-policy Soft Actor-Critic (SAC) (Haarnoja et al., 2018). For differentiable RL baselines ($\partial$RLs), we employ SHAC and AHAC, which, to the best of our knowledge, represent the SOTA in this category. For each environment and baseline, we used the tuned hyperparameters from Georgiev et al. (2024).

**Metric.** We evaluate performance in terms of the collected LTL rewards averaged over 5 seeds since they can serve as proxies for satisfaction probabilities. We considered two criteria: (1) the maximum return achieved and (2) the speed of convergence. To maintain consistency, we used differentiable LTL rewards across all baselines as, for non-differentiable baselines, we observed no performance difference between the differentiable and discrete LTL rewards.

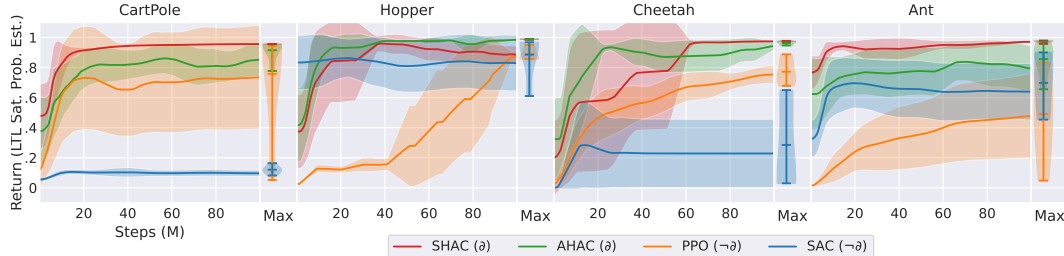

Figure 2: **Comparison Across Environments: Differentiable vs. Discrete LTL Rewards.** The wider plots show the learning curves of all baseline algorithms, while the narrower plots on the right display the maximum returns achieved after 100 M steps. All results are averaged over 5 random seeds, and the curves are smoothed using max and uniform filters for visual clarity. The reported returns, bounded between 0 and 1, serve as proxies for the probability of satisfying the LTL specifications. In all the environments, algorithms utilizing **differentiable** LTL rewards (SHAC, AHAC) rapidly learn near-optimal policies, whereas those relying on discrete LTL rewards (PPO, SAC), display high variance, converge slowly, or are stuck with sub-optimal policies.

**CartPole.** The CartPole environment consists of a cart that moves along a one-dimensional track, with a pole hinged to its top that can be freely rotated by applying torque. The system yields a 5-dimensional observation space and a 1-dimensional action space. The control objective is to move the tip of the pole through a sequence of target positions while maintaining the cart within a desired region as much as possible and ensuring the velocity of the cart always remains within safe boundaries. We capture these requirements in LTL as follows:

$$\varphi_{\text{cartpole}} = \underbrace{\square\text{`}|\texttt{cart\_vx}|<v_0\text{'}}_{\text{safety}} \wedge \underbrace{\square\Diamond\text{`}|\texttt{cart\_x}|<x_0\text{'}}_{\text{repetition}} \wedge \underbrace{\Diamond\big(\text{`}|\texttt{pole\_z-}z_0|<\Delta\text{'}\wedge\Diamond\text{`}|\texttt{pole\_z-}z_1|<\Delta\text{'}\big)}_{\text{reachability \& sequencing}}.$$

Here, $\texttt{cart\_x}$, $\texttt{cart\_vx}$, and $\texttt{pole\_z}$ represent the cart position, the cart velocity, and the pole height respectively. This formula demonstrates how LTL can be leveraged to encode both complex safety constraints and performance objectives. We set $x_0 = 10$ m, $v_0 = 10$ m/s as boundaries, $z_0 = -1$ m, $z_1 = 1$ m as the target positions, and $\Delta = 25$ cm as the allowable deviation.

**Legged Robots.** We consider three legged-robot environments: Hopper, Cheetah, and Ant. The Hopper environment features a one-legged robot with 4 components and 3 joints, resulting in a 10-dimensional state space and a 3-dimensional action space. The Cheetah environment consists of a two-legged robot with 8 components and 6 joints, yielding a 17-dimensional state space and a 6-dimensional action space. The Ant environment includes a four-legged robot with 9 components and 8 joints, producing a 37-dimensional state space and an 8-dimensional action space. In all three environments, the control task requires always keeping the torso/tip of the robot above a critical safety height, maintaining a certain distance between the torso/tip and the critical height as often as possible, and accelerating the robot forward, and then bringing the robot to a full stop. We formalize this task in LTL as follows:

$$\varphi_{\text{legged}} = \underbrace{\square\text{`}\texttt{torso\_z>}z_0\text{'}}_{\text{safety}} \wedge \underbrace{\square\Diamond\text{`}\texttt{torso\_z>}z_1\text{'}}_{\text{repetition}} \wedge \underbrace{\Diamond\big(\text{`}\texttt{torso\_vx>}v_1\text{'}\wedge\Diamond\text{`}\texttt{torso\_vx<}v_0\text{'}\big)}_{\text{reachability \& sequencing}}. \tag{10}$$

Here, $\texttt{torso\_z}$ and $\texttt{torso\_vx}$ denote the height and horizontal velocity of the robots. This formula captures several key aspects of LTL, including, safety, reachability, sequencing, and repetition. The values of $z_0$ and $z_1$ were chosen based on the torso height of each robot in their referential system. Specifically, we used $z_0 = -110$ cm, $z_1 = -105$ cm for Hopper; $z_0 = -75$ cm, $z_1 = -70$ cm for Cheetah; and $z_0 = 0$ cm, $z_1 = 5$ cm for Ant, where $z_0$ denotes the critical safety height and $z_1$ represents a safe margin above it. We set $v_1 = 1$ m/s, $v_1 = 3$ m/s, and $v_1 = 1.5$ m/s for Hopper, Cheetah, and Ant, respectively, reflecting movement speeds relatively challenging yet achievable for each of the robots. For deceleration, we set $v_0 = 0$ m/s for all the environments. An illustration of a policy learned from this specification for Cheetah is provided in Fig. 4 in Appx. F.

**Results.** Fig. 2 presents our simulation results. Across all environments, $\partial$RL algorithms that leverage our differentiable LTL rewards consistently outperform $\not\partial$RL algorithms in terms of both maximum return achieved and learning speed from the LTL specifications.

*CartPole.* The safety specification induces an automaton with three states, each having 64 transitions–but only one of these transitions yields a reward. This extreme sparsity, even in a low-dimensional state space, severely hinders the learning process for $\not\partial$RLs, as shown in the leftmost plot of Fig. 2. In contrast, $\partial$RL algorithms leverage the gradients provided by differentiable rewards, enabling them to efficiently learn policies that nearly satisfy the LTL specification. Specifically, $\partial$RLs converge to near-optimal policies (Pr>0.8) within just 20 M steps, whereas $\not\partial$RLs (SAC: all seeds; PPO: one seed) fail to learn any policy that achieves meaningful reward, even after 100 M steps.

*Legged Robots.* As we move to environments with higher-dimensional state spaces–10, 17, and 37 dimensions for Hopper, Cheetah, and Ant, respectively–even relatively simple LTL specifications pose a significant challenge for $\partial\!\!\!/$RLs. The automata derived from the LTL specifications in these environments consist of four states, each with 16 transitions, of which four transitions in the third state yield rewards. Reaching this state, however, requires extensive blind exploration of the state space, making it significantly harder for $\partial\!\!\!/$RLs to learn optimal control policies. On the other hand, $\partial$RLs, guided by LTL reward gradients, quickly identify high-reward regions of the state space and learn effective policies.

For Hopper, $\partial$RLs converge to near-optimal policies (Pr>0.8) within 20 M steps, while PPO requires the full 100 M steps to converge, and one SAC seed gets trapped in a local optimum. For Cheetah, $\partial$RLs attain optimal performance (Pr>0.9), whereas PPO converges to a suboptimal policy even after 100 M steps, and SAC consistently fails by getting stuck in poor local optima. For Ant, $\partial$RLs again learn near-optimal policies rapidly, while $\partial\!\!\!/$RLs converge only to suboptimal policies.

**Generalization to Reward Machines.** Our approach renders automaton-based rewards differentiable and can therefore be readily applied to frameworks such as RMs. We conducted analogous experiments in the Cheetah environment described in Icarte et al. (2022). Specifically, we used the same RMs from TASK 1 and TASK 2 and made them differentiable with our method. We then trained policies using the SHAC algorithm. Table 1 in Appx. G reports the returns obtained with the differentiable RMs alongside the best returns reported in Figure 10 of Icarte et al. (2022). Our differentiable RM-based approach significantly outperforms all discrete RM baselines.

**Ablation Study.** To isolate the impact of LTL reward differentiability from inherent algorithm and environment properties, we conducted two ablation studies. First, we trained SHAC and AHAC with discrete LTL rewards. The corresponding learning curves are shown in Fig. 7 in Appx. H. We observe a substantial performance degradation across all environments: the performance of SHAC/AHAC with discrete LTL rewards is markedly lower than with differentiable ones, and is mostly lower than that of PPO and SAC. The performance drop is especially pronounced in the higher-dimensional environments, Cheetah and Ant, where no reasonable policy is obtained. This suggests that the strong performance of $\partial$RLs in Fig. 2 is primarily due to our differentiable LTL rewards. Second, we trained each approach under simplified versions of the LTL formulas ((21) in Appx. H) lacking the complexity that makes learning from LTL challenging. Each algorithm eventually learns an optimal policy (Pr $> 0.9$) for all environments (see Fig. 8 in Appx. H). These results support the conclusion that the lower performance of $\partial\!\!\!/$RLs in Fig. 2 arises primarily from their inability to handle complex LTL formulas as effectively as $\partial$RLs with our differentiable rewards, rather than from environment-specific properties.

## 6 CONCLUSION

In this work, we tackle the challenge of scalable RL for temporally extended and formally specified tasks. By adopting LTL as our specification framework, we ensure objective correctness and avoid the reward-misspecification issues common in conventional RL. To overcome the learning inefficiencies caused by sparse logical rewards, we introduce a method that leverages differentiable simulators, enabling gradient-based learning directly from LTL objectives without compromising expressiveness or correctness. Our approach employs soft-labeling techniques that preserve differentiability through the transitions of automata derived from LTL formulas, yielding an end-to-end differentiable learning framework. Across simulated experiments, we show that our framework substantially accelerates learning compared to state-of-the-art non-differentiable baselines, pointing toward more reliable and scalable deployment of autonomous systems in real-world environments.

Our approach accelerates learning from LTL specifications by leveraging differentiable RL algorithms and gradients provided by differentiable simulators. Consequently, the overall performance of our method is inherently tied to the quality and efficiency of the underlying simulators and RL algorithms, and cannot be directly extended to discrete MDPs. A further consideration is that our method introduces an additional hyperparameter, the activation function used for probability estimation, which should be tuned for optimal performance. Another challenge lies in the formalization of LTL specifications: while LTL offers a more intuitive and structured way to specify tasks than manual reward engineering, it still requires familiarity with formal logic and sufficient domain knowledge to define meaningful bounds. An immediate direction for future work is to combine our approach with other techniques such as counterfactual experience replay and barrier nets. Another direction is to design LTL-specific differentiable RL methods that exploit the compositional structure of the automata to enable efficient exploration and transfer. See Appx. J for further discussion.

## ETHICS STATEMENT

This work complies with the ICLR Code of Ethics. All experiments in this study were conducted exclusively in simulated environments; therefore, no human participants, sensitive data, or real-world deployments were involved, thereby eliminating ethical risks associated with those settings.

## REPRODUCIBILITY STATEMENT

The source code needed to reproduce the results reported in this manuscript is included in the supplementary material, together with a `README.md` explaining the required steps. Upon acceptance, we will make the complete source code publicly available.

## ACKNOWLEDGMENTS

This work is supported in part by NSF-EFRI #2422282, IARPA HAYSTAC, Brendan Iribe Endowed Professorship, Dr. Barry Mersky and Capital One E-Nnovte Endowed Professorships, University of Maryland Distinguished University Professorship, and ARL-UMD ArtIAMAS Cooperative Agreement. We sincerely thank all the reviewers for their detailed and constructive feedback.

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

## A    REINFORCEMENT LEARNING WITH TEMPORAL LOGICS

**Deep RL.**    The rapid progress in machine learning during the 2010s, particularly in deep learning (LeCun et al., 2015; Goodfellow et al., 2016; Arulkumaran et al., 2017), facilitated by improved neural architectures and enhanced computational capabilities, significantly advanced deep reinforcement learning (RL). This enabled solving complex, high-dimensional, and long-horizon sequential decision-making tasks previously considered infeasible (Mnih et al., 2015; Silver et al., 2016; 2017; 2018; Vinyals et al., 2019). Such achievements have attracted growing interest from the control and robotics communities. Although deep RL has been effectively employed in structured, task-specific robotic and autonomous systems (Kober et al., 2013; Levine et al., 2016; Chen et al., 2017; Cui et al., 2017; Liu et al., 2018; Lu et al., 2018; Hwangbo et al., 2019; Cui et al., 2019; Peng & Shen, 2020; Andrychowicz et al., 2020; Lee et al., 2020; Yu et al., 2021; Kiran et al., 2021; Aradi, 2022), significant concerns around safety and reliability remain. In response, researchers have explored integrating RL with formal methods, particularly temporal logics, to enhance system reliability and verification.

**Model-Based RL for LTL.**    Initial attempts to combine linear temporal logic (LTL) with RL (Fu & Topcu, 2014b; Brázdil et al., 2014; Wen & Topcu, 2016) relied on model-based approaches. These methods required precise knowledge of the transition structure of the underlying Markov decision processes (MDPs) to precompute accepting components of a product MDP constructed using a Deterministic Rabin Automaton (DRA) based on LTL specifications. This approach transformed satisfying temporal logic constraints into reachability problems solvable via RL. Despite providing probably approximately correct (PAC) guarantees, the complexity and frequent unavailability of accurate transition models limit their applicability, especially in deep RL contexts.

**Model-Free RL for LTL.**    To address these limitations, model-free RL methods emerged, such as reward machines (RMs) (Toro Icarte et al., 2018; Icarte et al., 2018; Camacho et al., 2019; Icarte et al., 2022) for co-safe LTL and $LTL_f$ fragments, which directly generate rewards from the acceptance conditions of automata derived without explicit knowledge of transition dynamics. The introduction of limit-deterministic Büchi automata (LDBAs) (Hahn et al., 2015; Sickert et al., 2016), simplifying model checking by utilizing simpler Büchi conditions instead of Rabin conditions, further facilitated structured reward design (Hasanbeig et al., 2019; 2023). These advancements also resulted in improved correctness (Hahn et al., 2019) and stronger convergence guarantees (Bozkurt et al., 2020a).

**STL and Other Temporal Logics.**    Researchers have also explored alternative continuous-time temporal logics to generate informative reward signals such as STL and truncated LTL (TLTL) (Li et al., 2017; 2019). STL and TLTL allow robustness scores to serve as rewards in finite-horizon tasks (Aksaray et al., 2016). However, these scores typically depend on historical information, violating the Markov assumption and restricting their use in long-horizon, stochastic, or value-based RL settings. We note that the syntax of our differentiable LTL formalism coincides with TLTL as well as the STL fragment without time constraints; however, the semantics is defined based on discrete-time observations, which allows for compact automaton construction rather than robustness scores, yielding an efficient memory mechanism.

**Extended RL for LTL.**    Building on successful applications of LTL-based rewards in RL, researchers extended these methodologies into broader domains. These include stochastic games (Hahn et al., 2020; Bozkurt et al., 2021b; 2024), modular deep RL frameworks (Cai et al., 2021a; Jackermeier & Abate, 2025), reinforcement learning under workspace uncertainties (Cai et al., 2021b), secure planning against stealthy adversaries (Bozkurt et al., 2021a; Cui et al., 2023), learning within cluttered environments (Cai et al., 2023a). Additionally, recent developments include heuristic-driven learning (Kantaros, 2022), policy optimization strategies (Voloshin et al., 2022), quantum-based action spaces (Cai et al., 2023b), experience replay enhancements (Voloshin et al., 2023), transformer models (Tian et al., 2023), handling partially known semantics (Verginis et al., 2024), average reward formulations (Le et al., 2024), PAC guarantees (Perez et al., 2024), and goal-conditioned LTL-RL (Yalcinkaya et al., 2024). Theoretical analyses have explored computational intractability (Yang et al., 2022), discounting sensitivity (Xuan et al., 2024), and convergence properties (Shao & Kwiatkowska, 2023).

## B  PATHS AND POLICIES

**MDP Paths.**  An MDP begins in an initial state sampled from the initial state distribution $s_0 \sim p_0(\cdot)$ and evolves by transitioning from a current state $s$ to a next state $s'$ through an action $a$, as determined by the transition function: $s' = f(s, a)$. In each state $s$, the policy observes a set of atomic propositions provided by the labeling function $L(s)$. The sequence of visited states is called a path and is formally defined below:

**Definition 3.** *A path of an MDP $M$ is defined as an infinite sequence of states $\sigma = s_0 s_1 \ldots$, where each $s_i \in S$, such that $p_0(s_0) > 0$ and for every $t > 0$, there exists an action $a_t \in A$ with $f(s_t, a_t) = s_{t+1}$. We denote the $t$-th state in the sequence as $\sigma[t]$, the prefix up to $t$ as $\sigma[:t] = s_0 s_1 \ldots s_t$, and the suffix starting from $t + 1$ as $\sigma[t+1:] = s_{t+1} s_{t+2} \ldots$. The corresponding sequence of labels for $\sigma$ is referred to as the trace, defined by $L(\sigma) := L(s_0) L(s_1) \ldots$.*

**Finite-Memory Policies**

**Definition 4.** *A finite-memory policy for an MDP $M$ is defined as a tuple $\pi = (\mathfrak{M}, \mathfrak{m}_0, \mathfrak{T}, \mathfrak{a})$, where:*

- *$\mathfrak{M}$ is a finite set of modes (memory states);*

- *$\mathfrak{m}_0 \in \mathfrak{M}$ is the initial mode;*

- *$\mathfrak{T} : \mathfrak{M} \times S \times \mathfrak{M} \to [0, 1]$ is a probabilistic mode transition function such that for any current mode $\mathfrak{m}$ and state $s$, the probabilities over next modes sum to 1, i.e., $\sum_{\mathfrak{m}' \in \mathfrak{M}} \mathfrak{T}(\mathfrak{m}' \mid \mathfrak{m}, s) = 1$;*

- *$\mathfrak{a} : \mathfrak{M} \times S \times A \to [0, 1]$ is a probabilistic action selection function that assigns a probability to each action $a$ given the current mode $\mathfrak{m} \in \mathfrak{M}$ and state $s \in S$.*

A finite-memory policy acts as a finite-state machine that updates its internal mode (memory state) as states are observed, and specifies a distribution over actions based on both the current state and mode. The action at each step is thus selected according not only the current state but also the current memory state of the policy. In contrast to standard definitions of finite-memory policies (e.g., (Chatterjee & Henzinger, 2012; Baier & Katoen, 2008)), which typically assume deterministic mode transitions, this definition permits probabilistic transitions between modes. For further details, please refer to (Bozkurt, 2024).

## C  DIFFERENTIABLE REINFORCEMENT LEARNING

**Differentiable Simulators.**  Deep reinforcement learning (RL) provides a robust framework for learning control policies directly from high-dimensional, unstructured inputs without explicit human supervision. However, this flexibility introduces high sample complexity. To mitigate this, researchers developed methods such as distributed RL, massively parallel GPU-based RL, and model-based RL approaches. Recently, there has been significant interest in accelerating model-based RL using differentiable simulators, which enable gradient-based optimization by analytically or automatically computing gradients of states and rewards with respect to actions (Carpentier & Mansard, 2018; Geilinger et al., 2020; Qiao et al., 2021; Xu et al., 2021; Werling et al., 2021; Heiden et al., 2021; Freeman et al., 2021). These simulators can be represented as differentiable transition functions $s_{t+1} = f(s_t, a_t)$, where $s_t$ and $a_t$ represent the state and action at time step $t$, respectively, and $s_{t+1}$ is the next state at time step $t + 1$. In the context of reinforcement learning, a common choice for the differentiable loss function is the negative of the return, defined as the sum of discounted rewards: $\mathcal{L} = -G_H(\sigma) = -\sum_{t=0}^{H} \gamma_t r_t$, where $H$ is the time horizon, $\sigma = s_0 s_1 \ldots$ is the trajectory, $r_t$ is the reward at time step $t$, and $\gamma_t$ is the discount factor applied at that step. The backward pass then computes the gradients as follows:

$$\frac{\partial G_H}{\partial s_t} = \frac{\partial G_H}{\partial s_{t+1}} \frac{\partial f}{\partial s_t}, \qquad \frac{\partial G_H}{\partial a_t} = \frac{\partial G_H}{\partial s_{t+1}} \frac{\partial f}{\partial a_t}. \tag{11}$$

By chaining these gradients, the optimization updates propagate effectively throughout trajectories.

**Policy Gradient Objective.**  In policy gradient RL over a finite horizon $H$, the goal is to find optimal parameters $\theta^* = \operatorname{argmax}_\theta J(\theta)$, such that:

$$J(\theta) = \mathbb{E}_{\sigma \sim M_{\pi_\theta}}[G_H(\sigma)], \tag{12}$$

where $\sigma \sim M_{\pi_\theta}$ denotes the random trajectory $\sigma$ drawn from the Markov chain $M_{\pi_\theta}$ induced by the policy $\pi_\theta$ parameterized by $\theta$. If a differentiable model is available, optimization can leverage first-order gradients:

$$\nabla_\theta^{[1]} J(\theta) = \mathbb{E}_{\sigma \sim M_{\pi_\theta}}[\nabla_\theta G_H(\sigma)], \tag{13}$$

or employ model-free zeroth-order gradients via the policy gradient theorem:

$$\nabla_\theta^{[0]} J(\theta) = \mathbb{E}_{\sigma \sim M_{\pi_\theta}}\left[ G_H(\sigma) \sum_{t=0}^{H-1} \nabla_\theta \log \pi_\theta(a_t|s_t) \right]. \tag{14}$$

Both gradients can be approximated through Monte Carlo sampling:

$$\hat{\nabla}_\theta^{[1]} J(\theta) = \frac{1}{N} \sum_{i=1}^{N} \nabla_\theta G_H(\sigma^{(i)}), \tag{15}$$

$$\hat{\nabla}_\theta^{[0]} J(\theta) = \frac{1}{N} \sum_{i=1}^{N} G_H(\sigma^{(i)}) \sum_{t=0}^{H-1} \nabla_\theta \log \pi_\theta(a_t^{(i)}|s_t^{(i)}). \tag{16}$$

## D $\omega$-AUTOMATA

An LTL formula $\varphi$ can be translated into a finite-state automaton that operates over infinite paths, known as an $\omega$-automaton. We denote the automaton corresponding to a specific formula $\varphi$ by $\mathcal{A}\varphi$. An $\omega$-automaton $\mathcal{A}\varphi$ accepts a path $\sigma$ if and only if $\sigma \models \varphi$, and rejects it otherwise. We begin by formally introducing a general type of $\omega$-automaton, called a nondeterministic Rabin automaton (NRA), which can be systematically derived from any LTL formula (Baier & Katoen, 2008). We then focus on specific subclasses of NRAs that are particularly relevant to our work.

**Definition 5.** *A nondeterministic Rabin automaton (NRA) derived from an LTL formula $\varphi$ is defined as a tuple $\mathcal{A}\varphi = (Q, q_0, \Sigma, \delta, \text{Acc})$, where:*

- *$Q$ is a finite set of automaton states;*

- *$q_0 \in Q$ is the initial automaton state;*

- *$\Sigma = 2^{\mathtt{A}}$ is the input alphabet, where $\mathtt{A}$ is the set of atomic propositions;*

- *$\delta : Q \times (\Sigma \cup \{\varepsilon\}) \to 2^Q$ is the transition function, which is complete and deterministic on $\Sigma$ (i.e., $|\delta(q, w)| = 1$ for any $q \in Q$ and $w \in \Sigma$), but may include nondeterministic $\varepsilon$-transitions (i.e., it is possible that $|\delta(q, \varepsilon)| = 0$ or $|\delta(q, \varepsilon)| > 1$);*

- *Acc is a set of $k$ accepting pairs $\{(B_i, C_i)\}_{i=1}^k$ where $B_i, C_i \subseteq Q$ for $i \in \{1, \ldots, k\}$.*

Given an infinite word $\omega = w_0 w_1 \ldots$, a *run* of the automaton is a sequence of transitions $\tau_\omega = (q_0, w_0, q_1), (q_1, w_1, q_2), \ldots$ such that $q_{t+1} \in \delta(q_t, w_t)$ for all $t \geq 0$. The word $\omega$ is *accepted* if there exists such a run and at least one accepting pair $(B_t, C_t)$ satisfying the Rabin condition: the run visits some states in $B_t$ infinitely often and all states in $C_t$ only finitely often. This acceptance condition is known as the Rabin condition, which can be formalized as $\omega \models \varphi \iff \exists t : \text{Inf}(\tau_\omega) \cap B_i \neq \varnothing \wedge \text{Inf}(\tau_\omega) \cap C_i = \varnothing$ where $\text{Inf}(\tau_\omega)$ denotes the set of states visited infinitely many times in the run $\tau_\omega$. Similarly, a path $\sigma$ is considered *accepted* by the automaton if its trace $L(\sigma)$ forms a word accepted by $\mathcal{A}_\varphi$.

We focus on a specific subclass of NRAs known as limit-deterministic Büchi automata (LDBAs), which feature a simplified acceptance criterion. Despite their reduced complexity, LDBAs retain the full expressive power of general NRAs and can be systematically derived from LTL formulas (Hahn et al., 2015; Sickert et al., 2016).

**Definition 6.** *An LDBA is an NRA defined as $\mathcal{A}_\varphi = (Q, q_0, \Sigma, \delta, \text{Acc})$ that satisfies the following:*

- *The acceptance condition consists of a single pair with an empty second set, i.e., $\text{Acc} = (B, \varnothing)$, meaning that the run must visit some states in $B$ infinitely often. This is known as the Büchi condition, and for simplicity, we denote this acceptance condition by the set of accepting automaton states itself, i.e., $\text{Acc} = B$.*

- *The state space $Q$ is partitioned into two disjoint subsets: an initial component $Q_I$ and an accepting component $Q_A$, satisfying:*

  - *All accepting states are contained in $Q_A$, i.e., $B \subseteq Q_A$;*
  - *All transitions within $Q_A$ are deterministic (no $\varepsilon$-transitions), i.e., for any $q \in Q_A$, $\delta(q, \varepsilon) = \varnothing$;*
  - *Transitions cannot go from $Q_A$ to $Q_I$, i.e., for all $q \in Q_A$ and $w \in \Sigma$, $\delta(q, w) \subseteq Q_A$.*

The defining feature of LDBAs is that accepting runs must eventually enter the accepting component $Q_A$ and remain there permanently. Once this transition occurs, all subsequent behavior is deterministic–this property is known as limit-determinism. LDBAs can be constructed such that every $\varepsilon$-transition leads directly into $Q_A$, which ensures at most one such transition occurs along any execution path. This construction, combined with limit-determinism, makes LDBAs particularly well-suited for quantitative model checking in MDPs (Sickert et al., 2016) unlike general nondeterministic automata. Thus, we assume all LDBAs under consideration possess this structure. More information can be found in (Bozkurt, 2024).

## E    PARKING EXAMPLE: DIFFERENTIABLE RL VS NON-DIFFERENTIABLE RL

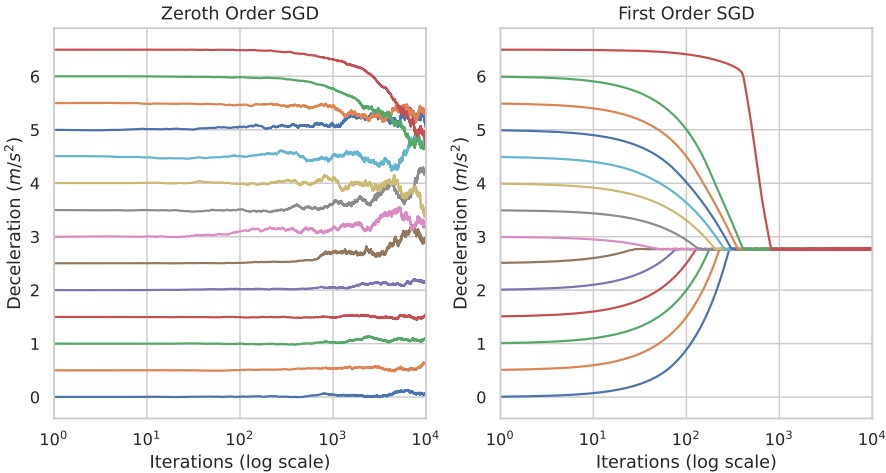

Figure 3: Convergence speed comparison of stochastic gradient descent algorithms using $\bar{\nabla}_{\theta}^{[0]}$ and $\bar{\nabla}_{\theta}^{[1]}$ for the parking example ($N = 10$).

## F    LTL SPECIFICATIONS AND AUTOMATA

**LTL Specifications used in the Experiments.**

- CartPole:

```
G("position_x>-10" & "position_x<10") & G("velocity_x>-10.0" & "velocity_x<10.0")
& F("cos_theta<-0.5" & F"cos_theta>0.5")
```

- Hopper:

```
G"torso_height>-11.0" & GF"torso_height>-10.5" & F("torso_velocity_x>1.0" & F"torso_velocity_x<0")
```

- Cheetah:

```
G"tip_height>-7.5" & GF"tip_height>-7.0" & F("tip_velocity_x>3.0" & F"tip_velocity_x<0")
```

- Ant:

```
G"torso_height>0.0" & GF"torso_height>0.5" & F("torso_velocity_x>1.5" & F"torso_velocity_x<0")
```

**Illustration of the LTL Specification for Cheetah.**

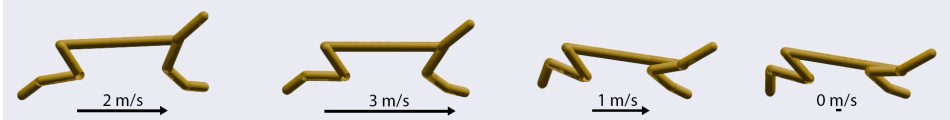

Figure 4: **Task Specification with LTL.** This figure illustrates a Cheetah policy learned by SHAC using differentiable rewards derived via our approach from the LTL formula $\varphi_{\text{legged}}$, which specifies accelerating forward, stopping, and maintaining a safe tip-to-ground distance. Specifying the desired behaviors of robots using the high-level language LTL provides is an intuitive alternative to manually designing reward functions, which often require extensive domain expertise and risk unintended behaviors. Enabling learning directly from LTL unlocks new possibilities for robust, safe, and flexible robotic applications. See the supplementary material for the video.

**Automata Derived from LTL Specifications for CartPole and Ant.**

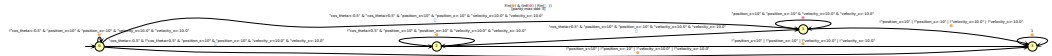

Figure 5: The $\omega$-automaton derived from $\varphi_{\text{cartpole}}$.

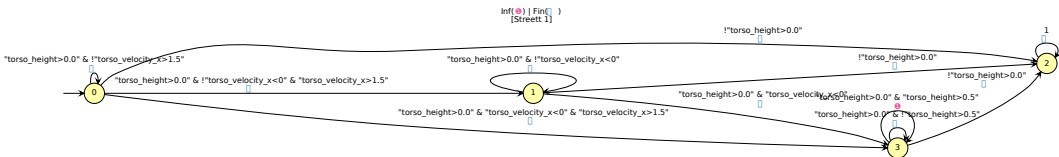

Figure 6: The $\omega$-automaton derived from $\varphi_{\text{legged}}$ for the Ant environment.

**Other example LTL formulas.** LTL can be used to specify temporal properties of a wide range of robotics tasks:

- safety; e.g., "always avoid obstacles and remain below the joint angle and velocity thresholds":

$$\varphi = \Box\Big( \bigwedge_{\text{obstacle}_i} \text{dist\_to\_obstacle}_i > 0 \bigwedge_{\text{joint\_angle}_i} \text{joint\_angle}_i < \beta_{\max} \bigwedge_{\text{joint\_vel}_i} \text{joint\_vel}_i < \dot{\beta}_{\max} \Big); \qquad (17)$$

- reachability; e.g.; "accelerate to a target velocity in x direction":

$$\varphi = \Diamond\text{torso\_vel}_x > v_{\text{target}}; \qquad (18)$$

- sequencing; e.g.; "move to the position 1, then move to the position 2, and after that move to the position 3":

$$\varphi = \Diamond\Big(\text{dist\_to\_pos}_1 < \delta_{\text{tol}} \wedge \Diamond\big(\text{dist\_to\_pos}_2 < \delta_{\text{tol}} \wedge \Diamond(\text{dist\_to\_pos}_3 < \delta_{\text{tol}})\big)\Big); \qquad (19)$$

- repetition; e.g.; "repeatedly monitor the region 1 and the region 2":

$$\varphi = \Box\Diamond\text{dist\_to\_region}_1 < \delta_{\text{tol}} \wedge \Box\Diamond\big(\text{dist\_to\_region}_2 < \delta_{\text{tol}}\big). \qquad (20)$$

## G    REWARD MACHINE GENERALIZATION RESULTS

Table 1: Comparison between differentiable RMs and discrete RMs for Cheetah.

|  | TASK 1 | | TASK 2 | |
| --- | --- | --- | --- | --- |
| Steps (K) | SHAC ($\partial$RL) | CRM ($\not\partial$RL) | SHAC ($\partial$RL) | HRM+RS ($\not\partial$RL) |
| 500 | $\mathbf{7.5} \pm 3.5$ K | $\approx 5.0 \pm 0.7$ K | $\mathbf{10.9} \pm 3.5$ K | $\approx 7.0 \pm 1.6$ K |
| 1000 | $\mathbf{12.2} \pm 1.9$ K | $\approx 7.0 \pm 0.4$ K | $\mathbf{16.6} \pm 1.4$ K | $\approx 8.1 \pm 1.9$ K |
| 1500 | $\mathbf{11.9} \pm 2.8$ K | $\approx 7.5 \pm 0.3$ K | $\mathbf{18.6} \pm 1.7$ K | $\approx 9.1 \pm 2.2$ K |
| 2000 | $\mathbf{13.7} \pm 3.2$ K | $\approx 8.0 \pm 0.4$ K | $\mathbf{19.4} \pm 1.9$ K | $\approx 9.1 \pm 2.0$ K |
| 2500 | $\mathbf{14.5} \pm 2.7$ K | $\approx 8.2 \pm 0.3$ K | $\mathbf{21.0} \pm 2.0$ K | $\approx 8.7 \pm 2.6$ K |
| 3000 | $\mathbf{15.4} \pm 2.5$ K | $\approx 8.3 \pm 0.3$ K | $\mathbf{21.1} \pm 1.9$ K | $\approx 9.1 \pm 2.8$ K |

## H    ABLATION STUDY RESULTS

We present ablation study results for differentiable LTL approaches in Figure 7 and Figure 8, respectively.

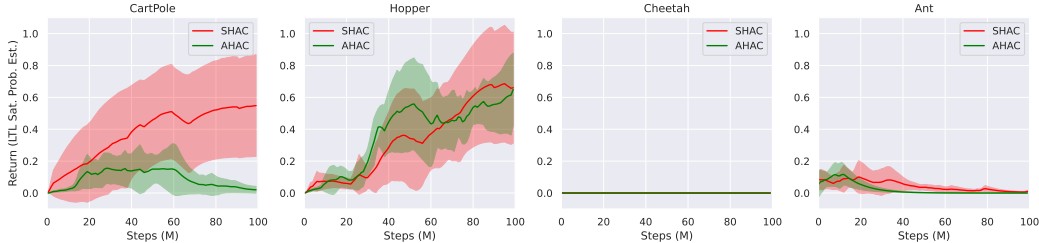

Figure 7: **Ablation study for differentiability of LTL rewards.** The maximum returns obtained after 100 M steps for SHAC and AHAC ($\partial$RLs) with discrete LTL rewards. Returns (0 to 1) indicate LTL satisfaction probabilities. With discrete LTL rewards, $\partial$RLs fail to learn near-optimal policies. However, as shown in Fig. 2, the *with our differentiable LTL rewards, they can successfully learn near-optimal policies.*

Simplified LTL formulas:
$$\varphi'_{\text{cartpole}} \coloneqq \Diamond \text{`}|\texttt{pole\_z-}z_0|\texttt{<}\Delta\text{`}, \qquad \varphi'_{\text{legged}} \coloneqq \Diamond \text{`}\texttt{torso\_vx>}v_1\text{`} \tag{21}$$
using $z_0 = -1$ m, $\Delta = 25$ cm for Cartpole, and $v_1 = 50$ cm/s for all the legged-robot environments.

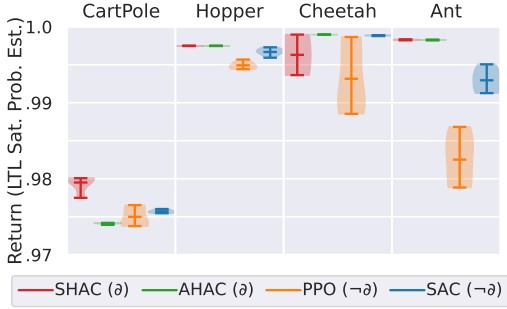

Figure 8: **Ablation study for complexity of LTL formulas.** The maximum returns obtained after 100 M steps for simplified LTL formulas (21). Returns (0 to 1) indicate LTL satisfaction probabilities. Under these simpler specifications, both $\not\partial$RLs and $\partial$RLs successfully learn near-optimal policies. However, as shown in Fig. 2, the performance of discrete $\not\partial$RLs degrades dramatically with increasing LTL complexity–*unlike differentiable $\partial$RLs, which maintain reasonable performance by leveraging the LTL rewards differentiability.*

# I   COMPARISON WITH OTHER LOGICS AND METHODS

## I.1   CONSTRAINED OPTIMIZATION

The main advantage of logical specifications over constrained optimization techniques such as barrier functions or model predictive control (MPC) is that they provide a general, unified, and arguably more intuitive language with high expressive power, which can be used to specify not only safety and liveness, but also memory-requiring properties such as sequencing and conditioning. These can be concatenated into a single objective formula using Boolean operators with well-defined probabilistic semantics. We note that these constrained optimization techniques might handle safety properties more effectively in a differentiable setting. However, we emphasize that these techniques are not mutually exclusive with our approach can be incorporated in a setting where safety constraints are prioritized over other specifications and handled by such techniques.

## I.2   SIGNAL TEMPORAL LOGIC

**History-Dependence.**   Temporal logics such as signal temporal logic (STL) are designed for continuous state variables and time constraints and thereby, generally, more compatible with differentiable systems. The main advantage of LTL, however, over these logics, is the compact memory mechanism provided as an automaton. Evaluating STL satisfaction or robustness, therefore, requires access to the full trajectory, as these metrics can only be computed after the trajectory ends. This necessitates storing entire trajectory histories, which directly violates the Markovian assumption fundamental to value-based RL techniques. Unlike LTL, this issue cannot be resolved by augmenting the state space with a compact memory representation derived from automata. There are two common approaches to address this challenge. The first is to augment the state space with the full trajectory history, but this leads to prohibitively large and impractical state spaces for the longer horizon tasks we consider in our work. For instance, with a horizon of $H = 1024$ used in our experiments, the state space for the Ant environment would be of size $1024 \times 37 = 37,887$. The second approach involves applying policy optimization over the action history using backpropagation through time (BPTT). However, due to the well-known exploding and vanishing gradient problems, gradients quickly deteriorate and become unreliable beyond roughly 100 time steps. This issue is further exacerbated in stochastic environments, where optimization becomes ineffective even over a small number of steps. Using intermediate STL robustness scores, an approach sometimes adopted to address the sparse reward problem in practice, does not resolve this hard non-Markovian problem of STL specifications. In fact, optimizing for the sum of such intermediate rewards can lead to incorrect RL objectives, as it does not necessarily correspond to maximizing the satisfaction of the STL specification.

**Syntax and Semantics.**   We focus on a subset of LTL formulas where atomic propositions (APs) are defined as bounds on continuous signals (or their functions). This subset overlaps with the class of STL formulas that do not include real-time constraints. One advantage of these formulas is that we avoid the inherently discrete APs permitted by general LTL formulas, thereby ensuring differentiability of the system. For example, instead of relying on a discrete low-battery indicator (e.g., $a := \text{low\_battery}$), we define APs as continuous thresholds (e.g., $a := \text{battery\_level} < 20\%$). Another advantage is that abstracting away the real-time constraints inherent in general STL formulas enables us to employ an automata-based compact memory mechanism rather than storing the entire trajectory history, substantially improving learning efficiency. We note that although STL formulas without real-time constraints are syntactically equivalent to the LTL formulas we consider, STL differs significantly at the semantic level. Specifically, STL places greater emphasis on the quantitative robustness score computed over the entire trajectory and does not aim to construct compact, automata-based memory. This distinction also holds for other similar logics such as truncated LTL, which are syntactically equivalent but similarly focus on robustness scores.

## I.3   REWARD MACHINES

**Background.**   The starting point of reward machine research is the construction of automata from LTL formulas specified as RL tasks (Icarte et al. (2018)). Early work focused on the co-safe fragment of LTL Icarte et al. (2018) and LTL over finite traces (LTLf) Camacho et al. (2019), for which

a deterministic finite automaton (DFA) can be constructed. These DFAs were later termed reward machines (RMs), as their accepting states can be used to assign positive rewards. Approaches targeting general LTL specifications instead use limit-deterministic Büchi automata (LDBAs). While structurally similar to RMs, LDBAs differ in their acceptance condition, which is based on infinite visitation of accepting states (suitable for continuous tasks), and thus require a distinct discounting scheme for correctness and convergence. In this work, we focus on making automaton-based reward signals differentiable, including those from DFAs and therefore RMs. When dealing with LDBAs, our approach additionally handles non-deterministic transitions and infinite visitation objectives. Importantly, our method can be immediately applied, without modification, to any RM constructed from co-safe LTL or LTL$_f$ formulas using the atomic propositions defined in Section 2. As a result, our approach can directly accelerate learning with RMs. Furthermore, all existing techniques used with RMs, such as reward shaping Icarte et al. (2022), can also be integrated into differentiable RL. We note that while reward shaping does not affect the optimality of policies, it may lead to incorrect estimates of satisfaction probabilities.

**Discrete vs. Differentiable RMs.** Differentiable reward machines will outperform discrete ones due to their inherent ability to propagate gradients across time steps, which provides additional derivative information about future rewards and states with respect to current actions. However, we would like to emphasize two important limitations. First, the additional gradients are not always useful in practice, especially when they are backpropagated over a long horizon that may include sharp gradients caused by events such as collisions. There are illustrative examples of this issue in Suh et al. (2022). Therefore, the gradient flow should be carefully controlled by approaches, such as limiting the horizon (SHAC) or cutting off gradients at sharp collisions (AHAC). Second, differentiable environments or simulators are not always available. Although physical systems can usually be modeled with differentiable simulators, this is not always possible for systems with inherently discrete states, actions, or transition structures. In such cases, a discrete-to-continuous conversion or approximation might be derived on a case-by-case basis, with additional challenges.

## J  Limitations and Extensions

### J.1  Discrete MDPs

An extension of our approach to discrete MDPs may be possible under certain assumptions. For example, if the MDP is a discretized version of a continuous environment, then any continuous state can be described as a probability distribution over the neighboring discrete states. As another example, discrete actions such as on/off switches could be expressed as a probability of switching. However, our approach cannot be trivially extended to discrete MDPs with inherently discrete states, such as categorical variables.

### J.2  Stochastic Transition Functions

Our method is naturally compatible with stochastic environments. For consistency with existing differentiable simulators, we model the transition function as deterministic. Nevertheless, probabilistic differentiable transition dynamics can also be incorporated using techniques such as the reparameterization trick. In the current implementation, the initial state distribution is the only source of randomness.

### J.3  Differentiable State Labels

We note that assuming a differentiable underlying system is, in general, a strong requirement. However, we view our additional assumption that state labels are differentiable as relatively minor, since it largely follows from the standard assumption that state variables and transitions are differentiable in all differentiable-simulator-based approaches. For example, a label such as "the light is red" either (i) arises from a separate discrete transition system that must be fused with the simulator for the Markov assumption, thereby already violating differentiability, or (ii) is a discrete function of differentiable variables (e.g., positions, velocities, or time), in which case it can usually be smoothed with respect to those variables as in our approach. Thus, while differentiable state labels are in-

deed a strong assumption, this limitation mainly stems from the differentiability requirement on the simulator's transition function and is shared by all differentiable-simulator-based methods.

## J.4 Choosing $\beta$

The choice of $\beta$ can be critical, depending on the environment and task. From a theoretical perspective, a larger $\beta$ is preferable for the correctness of the objective, whereas from a practical perspective, a smaller $\beta$ generally improves convergence and stability. We therefore suggest starting with a smaller $\beta$ and gradually increasing it until the algorithm no longer converges or begins to exhibit stability issues. Domain knowledge about the environment and task can also help guide this choice. For example, if we expect reaching tasks or repetitive behaviors encoded in the LTL specifications to manifest within $H$ time steps, then $\beta$ can be chosen so that $\beta^H$ remains reasonably large (e.g., $> 0.05$).

## J.5 Exploiting Frequency of Visiting Accepting States

Tweaking the discount factors can indeed result in policies that are suboptimal in terms of LTL satisfaction. For example, if the discount factor is not large enough (leading to myopic behavior), a policy that visits accepting states frequently, say at every time step for the first 1000 steps and then never again, may be preferred over a policy that visits an accepting state once every 1000 steps indefinitely, even though the latter should be preferred in terms of formal LTL satisfaction. Therefore, we suggest choosing discount factors that are sufficiently large while still ensuring convergence and stability, as discussed above.

## J.6 Signal Functions

We assume that designing signal functions is part of requirement extraction and specification engineering, and therefore treat them as given in our approach. However, the signal functions can affect the performance of our approach. Specifically, gradients are backpropagated through the signal functions after the activation functions, and if the signal functions have sharp regions, they may produce gradients with very large magnitudes, which can in turn impede the learning updates. Therefore, smoother signal functions are preferable.

## J.7 Slope of Sigmoid Activation

The slope of the sigmoid function used in the labeling function can, in fact, be critical. A steeper slope induces gradients with large magnitudes around the boundary, which can affect stability, and gradients with small magnitudes away from the boundary, which can slow convergence. In contrast, a shallower slope can interfere with correctness. In practice, domain knowledge can significantly help in determining these parameters. For example, in our approach we considered lengths in centimeters, for which a standard sigmoid function mostly transitions from 0 to 1 within about 10 cm, which was suitable for our tasks, so we did not need to change the temperature parameter much. We believe, however, that a clipped version of the rectified linear unit, e.g., $y = \min(\max(\text{temp.} \cdot x + 0.5, 0), 1)$, would be a better choice, although we did not have time to experiment with such alternative activation functions and will certainly do so in future work.

## J.8 Differentiable Matrix Multiplication

We assume that the transition matrices are small enough to be stored in GPU VRAM, where differentiable matrix multiplications can be performed efficiently in parallel using available tools such as PyTorch or TensorFlow. However, if the transition matrix is too large to fit in VRAM; for example, due to very long LTL specifications involving many state labels, the computation is not trivial.

## J.9 Partial Differentiability

A very interesting, necessary, and exciting future research direction, as one of the reviewers of this paper indicated, is to extend our approach to the systems with partial differentiability. We believe that our approach cannot be readily extended, and that new RL algorithms are required for such

hybrid settings. One case to consider, however, is when the differentiable and discrete systems have their own separate states, actions, and transition systems; i.e., they are completely decoupled except through the rewards. In this case, it becomes possible to obtain derivatives of future rewards with respect to the actions of the differentiable system.

### J.10 COUNTERFACTUAL EXPERIENCE REPLAY AND REWARD SHAPING

Our approach can be extended to include counterfactual experience replay Voloshin et al. (2023) and reward shaping techniques Icarte et al. (2022) under some caveats. First, additional rewards provided by shaping should be a differentiable function of the state variables, and second, they should not compromise the formal objective of maximization of the satisfaction probabilities. The latter does not technically interfere with the learning algorithm, but violates the main goal of this paper to have a formally sound objective. Counterfactual experience replay can be integrated with the differentiable approaches SHAC and AHAC if the counterfactual trajectories are obtained by changing only the automaton states.

