# OpenReview forum: "Accelerated Learning with Linear Temporal Logic using Differentiable Simulation"
_ICLR.cc/2026/Conference — ICLR 2026 Poster_

### Official Review · Reviewer_h7aD · 2025-10-27

**Soundness:** 4
**Presentation:** 2
**Contribution:** 4
**Rating:** 8
**Confidence:** 4

**Summary:**

Logical formalisms enable the specification of agents’ behaviors and offer correct-by-construction objectives. In RL, reward design often struggles to capture the user’s intended task, resulting in misaligned or sparse signals. Linear temporal logic (LTL) provides an expressive language to specify such trajectory-level objectives but leads to discrete, sparse rewards that hinder policy optimization. This paper introduces an end-to-end differentiable framework that integrates LTL with differentiable simulators, allowing gradients to flow through both the environment and the logical specification. The approach replaces hard labeling with smooth probabilistic labeling, yielding differentiable transitions and rewards. The authors prove a theoretical bound linking discrete and differentiable LTL returns and empirically demonstrate faster learning and higher returns on continuous-control benchmarks. The authors also note the results hold in stochastic settings. The method also generalizes to reward machines, thus covering co-safe LTL and LTLf tasks. Overall, the framework bridges formal methods and deep RL through differentiable logic-based objectives.

**Strengths:**

- The proposed use of differentiable labeling functions and probabilistic automaton transitions provides a clear and elegant means of propagating gradients through LTL objectives.
- Theoretical contributions include a bound relating discrete and differentiable LTL rewards, ensuring that the relaxation remains faithful to the underlying specification.
- The algorithmic exposition is quite clear; gradients, $\epsilon$-actions, and update steps are explicitly shown in the final algorithm, supported by a detailed parking example that illustrates the difference between discrete and differentiable LTL rewards.
- The experimental evaluation is broad, covering easy-to-challenging continuous-control benchmarks (from 5-/1-D to 37-/8-D state-action spaces) and including ablation studies that isolate the role of differentiability.
- Comparisons between differentiable and discrete LTL baselines are convincing, showing consistent improvement in convergence and policy quality.
- The experiments on reward machines and their comparison to existing algorithms are particularly valuable, as they demonstrate compatibility across formal-specification frameworks and situate the contribution in a wider research context.

**Weaknesses:**

- While Theorem 2 mentions the applicability of the approach to stochastic settings, the environments presented in the paper are solely deterministic.
- The success of the approach largely relies on $\beta$, which allows deriving the sole reward signal the agent gets and serves as a second discount factor. $\beta$ is also critical in Theorem 2, providing a divergence between discrete and differentiable rewards. However, there is no discussion or intuition on how to choose $\beta$ in practice to achieve the theoretical guarantees.

*Remark*: I've had a hard time parsing the phrasing *"Further, the rewards are discounted less in non-accepting states to reflect that the number of visitations to non-accepting states are not important."* Since $\beta$ is a function of $\gamma$ and the ratio in Theorem 1 should converge to zero, I understand that $\beta$ should always be lower than $\gamma$, but the phrasing ("discounting less") is confusing.
- The signal functions $g_a$ underlying the soft labels remain heuristic and "hard-coded"; their design influence is not discussed.
- The paper is visually extremely dense, relying heavily on negative `\vspace` and compressed layout, which reduces readability.

**Questions:**

- How should we tune $\beta$ in practice?
- Could the approach be extended to environments that are only partially differentiable or hybrid, where discrete transitions coexist with differentiable dynamics?
- The proposed rewards depend on the frequency of visiting accepting states in the automaton. Could the authors clarify whether, by tweaking the discounts, agents might "exploit" repeated visits to accepting states to obtain high returns without maintaining long-term satisfaction of the LTL formula?
- How sensitive is the method to the slope of the sigmoid activation used in the labeling functions, and how does this affect learning stability or correctness? Did you consider enriching the sigmoid function with a temperature parameter in practice? Would it be a good idea?

---

> ### Author Response · Authors · 2025-11-26
>
> We are very thankful for your thoughtful feedback and constructive suggestions. We have addressed your questions and concerns in detail below and revised the manuscript accordingly. The discussed limitations are now included in the appendix and are referenced in the main text. We hope these revisions satisfactorily address your concerns.
>
> ## W1. (Extension to Stochastic Environments)
> Thank you for raising this point. Our method is naturally compatible with stochastic environments. For consistency with existing differentiable simulators, we model the transition function as deterministic. Nevertheless, probabilistic differentiable transition dynamics can also be incorporated using techniques such as the reparameterization trick. In the current implementation, the initial state distribution is the only source of randomness.
>
> ## W2/Q1. (Choosing Beta)
> We agree that the choice of $\beta$ can be critical, depending on the environment and task. From a theoretical perspective, a larger $\beta$ is preferable for the correctness of the objective, whereas from a practical perspective, a smaller $\beta$ generally improves convergence and stability. We therefore suggest starting with a smaller $\beta$ and gradually increasing it until the algorithm no longer converges or begins to exhibit stability issues. Domain knowledge about the environment and task can also help guide this choice. For example, if we expect reaching tasks or repetitive behaviors encoded in the LTL specifications to manifest within $H$ time steps, then $\beta$ can be chosen so that $\beta^H$ remains reasonably large (e.g., $> 0.05$).
>
> ## W3. (Remark on Discounting Less)
> We understand the phrase ``discounted less'' could be confusing, and we replaced it with "discounted with a larger factor".
>
> ## W4. (Signal Functions)
> We assume that designing signal functions is part of requirement extraction and specification engineering, and therefore treat them as given in our approach. However, we agree that signal functions can indeed affect the performance of our approach. Specifically, gradients are backpropagated through the signal functions after the activation functions, and if the signal functions have sharp regions, they may produce gradients with very large magnitudes, which can in turn impede the learning updates. Therefore, smoother signal functions are preferable.
>
> ## W5. (Readability)
> We agree that our paper is quite dense, and we will try to make it more concise and move some details to the appendix to make it more readable upon acceptance.
>
> ## Q2. (Partial Differentiability)
> This would be a very interesting, necessary, and exciting future research direction. We believe that our approach cannot be readily extended, and that new RL algorithms are required for such hybrid settings. One case to consider, however, is when the differentiable and discrete systems have their own separate states, actions, and transition systems; i.e., they are completely decoupled except through the rewards. In this case, it becomes possible to obtain derivatives of future rewards with respect to the actions of the differentiable system. We thank you for pointing out this future direction of many exciting possibilities.
>
> ## Q3. (Exploiting Frequency of Visiting Accepting States)
> Tweaking the discount factors can indeed result in policies that are suboptimal in terms of LTL satisfaction. For example, if the discount factor is not large enough (leading to myopic behavior), a policy that visits accepting states frequently, say at every time step for the first 1000 steps and then never again, may be preferred over a policy that visits an accepting state once every 1000 steps indefinitely, even though the latter should be preferred in terms of formal LTL satisfaction. Therefore, we suggest choosing discount factors that are sufficiently large while still ensuring convergence and stability, as discussed above.
>
> ## Q4. (Slope of Sigmoid Activation)
> The slope of the sigmoid function used in the labeling function can, in fact, be critical. A steeper slope induces gradients with large magnitudes around the boundary, which can affect stability, and gradients with small magnitudes away from the boundary, which can slow convergence. In contrast, a shallower slope can interfere with correctness. In practice, domain knowledge can significantly help in determining these parameters. For example, in our approach we considered lengths in centimeters, for which a standard sigmoid function mostly transitions from 0 to 1 within about 10 cm, which was suitable for our tasks, so we did not need to change the temperature parameter much. We believe, however, that a clipped version of the rectified linear unit, e.g., $y = \min(\max(\text{temp.}\cdot x+0.5, 0), 1)$, would be a better choice, although we did not have time to experiment with such alternative activation functions and will certainly do so in future work.

---

> > ### Comment · Reviewer_h7aD · 2025-11-27
> >
> > Thank you for addressing my comments and answering my questions.
> > Concerning the source of randomness, I had the re-parameterization trick in mind indeed; this means that your method may easily apply to classic, stochastic control tasks with Gaussian kernels as a source of stochasticity. Interesting.
> >
> > I am happy with the authors' answers, and I found the discussion very interesting. I hope that such details may at least enrich the Appendix of the final version.

---

### Official Review · Reviewer_QdnH · 2025-10-31

**Soundness:** 4
**Presentation:** 4
**Contribution:** 3
**Rating:** 4
**Confidence:** 4

**Summary:**

The paper introduces a method to make logic-based specifications of rewards or constraints differentiable. Classical logic-specification frameworks rely on a formula that defines objectives through the truth values of Boolean predicates, which are then translated into an automaton and combined with an MDP to form a product MDP. The problem is that the resulting automaton produces non-differentiable states, as these depend on discrete Boolean propositions. This paper proposes a relaxation by introducing _soft labels_ that yield probabilities instead of Boolean values. Consequently, the automaton is represented by _probability vectors_ rather than one-hot encodings, enabling (i) differentiable reward signals and (ii) reduced reward sparsity, a well-known issue in the literature. The also formally prove the discrepancy between the discrete and differentiable returns. They provide empirical evaluation.

**Strengths:**

- The proposed idea is interesting and their contribution naturally mitigates reward sparsity while maintaining differentiability, making it compatible with differentiable RL controllers.
- The mathematical formulation is sound and providing both differentiability and normalization.
- The proposed approach bridge the temporal abstraction provided by the LTL framework to the differentiable control settings.

**Weaknesses:**

- I believe that limiting the labels to continuous functions of the state is a strong assumption, that restricts the benefits of logic specifications to consider only predicates that are expressible as continuous functions. For instance, $a$ = "agent reached goal in $(x_g, y_g)$" is captured by your assumptions, becasue it can be expressed as a distance, which is a continuous function of the state $x$. Instead, $a$ = "the light is red" is a boolean and condition. This assumption limits the expressive power of the logical specification framework to consider only continuous labels.
- It would be nice to compare the performance of SHAC and AHAC with and without the LTL framework for a fair evaluation. Providing this comparison would clarify whether the observed improvements stem from the proposed LTL-based formulation or from the underlying algorithmic differences. Without this, is hard to tell what is the source of the improvement.

**Questions:**

- Regarding the claim on line 253 that "this computation can be efficiently done through differentiable matrix multiplication" can you comment on this? The transition matrix has size $|Q| \times |Q|$ do you assume that $|Q|$ is small in practice? Otherwise, the operation may not be computationally trivial.
- Can you elaborate on the point raised in weakness 1?
- Given that the framework can only express continuous functions of the state as labels, what are the advantages of relying on logical specifications rather than classical approaches such as Model Predictive Control, which naturally handles continuous constraints (aside from the temporal abstraction advantage)?

Typos:
- Line 122: "...The state space S (is) the..."
- Line 132: possible repetition of "the return."

---

> ### Author Response · Authors · 2025-11-26
>
> We are very thankful for your thoughtful feedback and constructive suggestions. We have addressed your questions and concerns in detail below and revised the manuscript accordingly. The discussed limitations are now included in the appendix and are referenced in the main text. We hope these revisions satisfactorily address your concerns.
>
> ## W1/Q1 (Label Differentiability Assumption):
> We agree that assuming a differentiable underlying system is, in general, a strong requirement. However, we view our additional assumption that state labels are differentiable as relatively minor, since it largely follows from the standard assumption that state variables and transitions are differentiable in all differentiable-simulator-based approaches. For example, a label such as "the light is red" either (i) arises from a separate discrete transition system that must be fused with the simulator for the Markov assumption, thereby already violating differentiability, or (ii) is a discrete function of differentiable variables (e.g., positions, velocities, or time), in which case it can usually be smoothed with respect to those variables as in our approach. Thus, while differentiable state labels are indeed a strong assumption, this limitation mainly stems from the differentiability requirement on the simulator’s transition function and is shared by all differentiable-simulator-based methods.
>
> ## W2 (SHAC/AHAC Comparison without Differentiable LTL Rewards):
> Thank you for suggesting this comparison; we belive adding such comparison improves the quality and the clarity of our contributions. We designed and conducted new experiments for this and updated the ablation study section of the manuscript. In order to have a fair comparison, we disabled the differentiability of our approach and trained the algorithms SHAC and AHAC with discrete rewards. Our results are shown in **Figure 7** (Appendix G) in the updated manuscript. We observe a substantial performance degradation across all environments: the performance of SHAC/AHAC with discrete LTL rewards is markedly lower than with our differentiable LTL rewards, and is mostly lower than that of PPO and SAC. The performance drop is especially pronounced in the higher-dimensional environments, Cheetah and Ant, where no reasonable policy is obtained.
>
> ## Q2 (Matrix Multiplication Efficiency):
> We assume that the transition matrices are small enough to be stored in GPU VRAM, where differentiable matrix multiplications can be performed efficiently in parallel using available tools such as PyTorch or TensorFlow. However, we agree that if the transition matrix is too large to fit in VRAM; for example, due to very long LTL specifications involving many state labels, the computation is not trivial, as indicated.
>
> ## Q3 (Comparison with Model Predictive Control):
> We believe that the main advantage of logical specifications over MPC is that they provide a general, unified, and arguably more intuitive language with high expressive power, which can be used to specify not only safety and liveness, but also memory-requiring properties such as sequencing and conditioning. These can be concatenated into a single objective formula using Boolean operators with well-defined probabilistic semantics. In contrast, MPC focuses on more constrained optimization with a fixed horizon, which, we agree, could be better integrated into RL settings for ensuring safety with continuous constraints. However, we emphasize that MPC is not mutually exclusive with our approach and can be combined with it in a setting where safety constraints are prioritized over other specifications and handled with MPC.
>
> ## Typos:
> Thank you for pointing out these typos; we fixed these and updated the manuscript accordingly.

---

### Official Review · Reviewer_kd6k · 2025-11-03

**Soundness:** 4
**Presentation:** 3
**Contribution:** 3
**Rating:** 6
**Confidence:** 3

**Summary:**

This paper introduces an end-to-end framework that combines Linear Temporal Logic (LTL) with differentiable simulators, allowing agents to learn directly from formal task specifications using gradient-based optimization. The authors argue that when LTL is used in reinforcement learning, it typically produces discrete and sparse rewards, which can slow down training and hurt performance. To address this, they “soften” the discrete automaton transitions through a soft-labeling technique, creating smooth, differentiable rewards while still maintaining the original task semantics. They also provide theoretical guarantees showing that the gap between the discrete LTL reward and the relaxed differentiable version is bounded under certain assumptions. Experiments compare their method against PPO and SAC agents trained with standard sparse LTL rewards, demonstrating improved performance.

**Strengths:**

The paper is clearly written and easy to understand. The authors do a nice job presenting their method.

The motivation is strong. LTL and other temporal-logic or formal-methods approaches often struggle with scalability and sparse rewards in RL, and this work tackles that important challenge.

The proposed approach appears sound. Using differentiable simulation creates a fully end-to-end training pipeline with gradient flow, which can speed up learning and potentially lead to better overall performance.

The authors include a theoretical analysis that bounds the difference between the original discrete reward and the relaxed differentiable version.

The experimental results are convincing and show promising improvements.

The related-work discussion is thorough and well-situated in the existing literature.

**Weaknesses:**

1. It is not fully clear how this method compares to other model-based RL approaches that also incorporate formal guarantees during training. For example, recent work that uses world models together with barrier certificates or STL can also be viewed as an end-to-end differentiable pipeline, since the world model acts as a differentiable simulator and the formal constraints guide policy learning. It would be helpful for the authors to discuss these connections more clearly and highlight the differences, including works like
Reference: State-Wise Safe Reinforcement Learning with Pixel Observations.

2. Unlike LTL, STL signals are already continuous rather than purely discrete. Why do the authors focus on making LTL differentiable instead of using STL, which may naturally fit differentiable learning?

3. The experiments do not include comparisons with model-based or STL-based RL baselines, which would help clarify the advantages of the proposed method.

**Questions:**

1. Page 3, the MDP formulation is more like a control formulation, generally we should have something like S x A x S -> Prob[0, 1] as transition dynamics.

---

> ### Author Response · Authors · 2025-11-26
>
> We thank you for your thoughtful feedback, constructive suggestions, and for drawing attention to key issues in our work. We have responded to your questions and concerns in detail below and revised the manuscript accordingly. Each of these limitations is now discussed in the appendix due to space constraints, while also being referenced in the main body of the paper. We hope our revisions and explanations adequately address your concerns.
>
> ## W1. (Comparison with Other Formal Approaches)
> Thank you for bringing this reference to our attention, we included this in our updated manuscript. We understand that some important details may have been obscured since we had to condense the related work section due to space limitations. The main advantage of LTL over constrained optimization techniques such as barrier functions is that LTL provides a unified, expressive, and often more intuitive language that can capture not only safety and liveness but also memory-requiring properties (e.g., sequencing and conditioning) within a single probabilistic objective. We note that these techniques might handle safety properties more effectively in a differentiable setting and can be incorporated into our approach to prioritize safety objectives over other temporal properties, as in existing work on STL with barrier functions. We discuss the advantages of LTL over STL below.
>
> ## W2. (Advantages of LTL over STL)
> We agree that other logics such as MTL or STL are designed for continuous state variables and time constraints and thereby, generally, more compatible with differentiable systems. The main advantage of LTL, however, over these logics, is the compact memory mechanism provided as an automaton. Evaluating STL satisfaction or robustness, therefore, requires access to the full trajectory, as these metrics can only be computed after the trajectory ends. This necessitates storing entire trajectory histories, which directly violates the Markovian assumption fundamental to value-based RL techniques. Unlike LTL, this issue cannot be resolved by augmenting the state space with a compact memory representation derived from automata. There are two common approaches to address this challenge. The first is to augment the state space with the full trajectory history, but this leads to prohibitively large and impractical state spaces for the longer horizon tasks we consider in our work. For instance, with a horizon of H=1024 used in our experiments, the state space for the Ant environment would be of size 1024 x 37 = 37,887. The second approach involves applying policy optimization over the action history using backpropagation through time (BPTT). However, due to the well-known exploding and vanishing gradient problems, gradients quickly deteriorate and become unreliable beyond roughly 100 time steps. This issue is further exacerbated in stochastic environments, where optimization becomes ineffective even over a small number of steps. Using intermediate STL robustness scores, an approach sometimes adopted to address the sparse reward problem in practice, does not resolve this hard non-Markovian problem of STL specifications. In fact, optimizing for the sum of such intermediate rewards can lead to incorrect RL objectives, as it does not necessarily correspond to maximizing the satisfaction of the STL specification. We hope our response provides sufficient information and clarification.
>
> ## W3. (STL Comparison)
> Due to this non-Markovian characteristic of STL (also see our response above), we were unable to obtain meaningful experimental results with STL for the tasks considered in our paper. We hope this clarification adequately explains why we did not use STL as a baseline.
>
> ## Q1. (Deterministic Transition Function)
> Thank you for pointing this out. For the sake of consistency with existing differentiable simulators, we define the transition function as deterministic. However, through approaches such as the reparameterization trick, a probabilistic differentiable transition function can also be accommodated. In our current approach, the only source of randomness comes from the initial state distribution.

---

> > ### Comment · Reviewer_kd6k · 2025-11-27
> >
> > Thanks for the response. The discussion above has sufficiently addressed my questions and concerns. I will increase my score to a 8.

---

### Official Review · Reviewer_Ay51 · 2025-11-03

**Soundness:** 3
**Presentation:** 3
**Contribution:** 3
**Rating:** 6
**Confidence:** 5

**Summary:**

This paper presents a logic-based reinforcement learning (RL) framework that introduces differentiable LTL rewardsby employing probabilistic, or “soft,” atomic propositions within differentiable MDP environments. The key idea is to integrate logic-based specification methods into differentiable RL, thereby bridging symbolic reasoning and gradient-based optimization. The authors further extend the framework to differentiable generalizations of reward machines, providing a smooth relaxation of discrete logical rewards. Theoretical results show that by appropriately choosing the *softening parameter* $\zeta$, the error between the discrete and differentiable logic-based RL formulations can be made arbitrarily small.

Empirical evaluations demonstrate that Short Horizon Actor-Critic (SHAC) and Adaptive Horizon Actor-Critic (AHAC) outperform discrete RL baselines such as PPO and SAC, confirming that the benefits of differentiable MDPs extend to logic-based rewards. The experiments also include a differentiable variant of reward machines, showing that differentiability continues to provide learning benefits even under structured quantitative objectives.

**Strengths:**

1. The integration of logical specifications into differentiable RL is conceptually elegant and addresses an important challenge in combining symbolic reasoning with continuous optimization.
2. The derivation of error bounds demonstrating convergence between discrete and differentiable logic-based RL formulations adds mathematical rigor to the approach.
3. Experiments with SHAC and AHAC validate the framework’s benefits over standard baselines, showing improvements consistent with expectations from differentiable modeling.
4. The paper clearly situates itself within state-of-the-art logic-based RL research and provides a helpful summary of related work on differentiable MDPs and reward machines.

**Weaknesses:**

1. It is unclear whether the proposed framework can be readily extended to discrete MDPs. Intuitively, a similar smooth approximation could be applied to the transition structure, but this is not discussed. Clarifying this would strengthen the paper’s generality.
2. The paper does not explore whether inherently discrete methods such as reward shaping or counterfactual reasoning could complement, or interfere with, the differentiable automata framework. Including experiments or discussion in this direction would add practical depth.
3. The paper focuses on LTL, but it would be helpful to discuss whether logics tailored to continuous systems, such as Metric Temporal Logic or Signal Temporal Logic, are more naturally suited to differentiable environments. A short comparison of their expressive and computational trade-offs would be helpful.

**Questions:**

1. Can the proposed approach be extended to discrete MDPs, perhaps via smooth approximations of the transition probabilities?
2. How would reward shaping or counterfactual experience replay interact with differentiable logical rewards?
3. Are continuous-time temporal logics such as MTL or STL more suitable for differentiable RL, and how might they affect interpretability or computational cost?
4. Discrete reward machines, with their inherently sparse reward structures, can sometimes perform competitively when combined with reward shaping or counterfactual experiences. Do the authors expect differentiable reward machines to consistently outperform discrete ones, or could sparsity occasionally confer advantages?

---

> ### Author Response · Authors · 2025-11-26
>
> We appreciate your thoughtful comments, constructive suggestions, and for highlighting key issues in our paper. We have addressed and clarified the questions and concerns as detailed below and have updated our manuscript accordingly. We included a discussion of each of these limitations in the appendix due to space constraints, while ensuring that they are also mentioned in the main body of the paper. We hope our responses provide sufficient information and clarification.
>
> ## W1/Q1 (Extension to Discrete MDPs):
> We believe that an extension to discrete MDPs, as suggested, may be possible under certain assumptions. For example, if the MDP is a discretized version of a continuous environment, then any continuous state can be described as a probability distribution over the neighboring discrete states. As another example, discrete actions such as on/off switches could be expressed as a probability of switching. However, our approach cannot be trivially extended to discrete MDPs with inherently discrete states, such as categorical variables.
>
> ## W2/Q2 (Reward Shaping and Counterfactual Experience Replay):
> Our approach can be extended to include reward shaping and counterfactual experience replay as suggested under some caveats. First, additional rewards provided by shaping should be a differentiable function of the state variables, and second, they should not compromise the formal objective of maximization of the satisfaction probabilities. The latter does not technically interfere with the learning algorithm, but violates the main goal of this paper to have a formally sound objective. Conterfactual experience replay can be integrated with the differentiable approaches SHAC and AHAC if the counterfactual trajectories are obtained by changing only the automaton states.
>
> ## W3/Q3 (Comparison with Logics for Continuous Systems):
> Thank you for pointing this out. We include a detailed comparison with MTL/STL in our response to Reviewer kd6K in W2 (“Advantages of LTL over STL”) and discuss the expressiveness, as well as syntactic and semantic differences, below.
>
> We focus on a subset of LTL formulas where atomic propositions (APs) are defined as bounds on continuous signals (or their functions). This subset overlaps with the class of STL formulas that do not include real-time constraints. One advantage of these formulas is that we avoid the inherently discrete APs permitted by general LTL formulas, thereby ensuring differentiability of the system. For example, instead of relying on a discrete low-battery indicator (e.g., $a := low\\_battery$), we define APs as continuous thresholds (e.g., $a := battery\\_level < 20\%$). Another advantage is that abstracting away the real-time constraints inherent in general STL formulas enables us to employ an automata-based compact memory mechanism rather than storing the entire trajectory history, substantially improving learning efficiency.
>
> Finally, we would like to emphasize that, although STL formulas without real-time constraints are syntactically equivalent to the LTL formulas we consider, STL differs significantly at the semantic level. Specifically, STL places greater emphasis on the quantitative robustness score computed over the entire trajectory and does not aim to construct compact, automata-based memory.
>
> ## Q4 (Discrete vs. Differentiable Reward Machines):
> We believe that, in general, differentiable reward machines will outperform discrete ones due to their inherent ability to propagate gradients across time steps, which provides additional derivative information about future rewards and states with respect to current actions. However, we would like to emphasize two important limitations. First, the additional gradients are not always useful in practice, especially when they are backpropagated over a long horizon that may include sharp gradients caused by events such as collisions. There are illustrative examples of this issue in Suh et al. (2022, “Do Differentiable Simulators Give Better Policy Gradients?”). Therefore, the gradient flow should be carefully controlled by approaches, such as limiting the horizon (SHAC) or cutting off gradients at sharp collisions (AHAC). Second, differentiable environments or simulators are not always available. Although physical systems can usually be modeled with differentiable simulators, this is not always possible for systems with inherently discrete states, actions, or transition structures. In such cases, a discrete-to-continuous conversion or approximation might be derived on a case-by-case basis, with additional challenges.
>
> In conclusion, we expect differentiable reward machines to outperform discrete ones, provided that the environment is differentiable and the gradient flow is controlled via carefully fine-tuned parameters such as the horizon or collision threshold.

---

### Author Response · Authors · 2025-12-04
**Author Final Remarks**

We sincerely thank the ACs for the extra effort they had to put in due to the complications caused by the leak issue. We thank the reviewers for their detailed and constructive feedback; addressing their concerns and engaging in discussions during the rebuttal helped us clarify and strengthen our contributions and improve the overall quality of the paper.

Our work presents the first end-to-end differentiable reinforcement learning (RL) framework for learning controllers from linear temporal logic (LTL) specifications in differentiable simulation environments, further narrowing the gap between formal methods and deep RL. We introduce a method that makes LTL-based automaton rewards differentiable in continuous control domains, enabling gradient-based optimization directly from formal specifications. Our approach significantly accelerates learning, achieving up to twice the returns of the baselines across diverse experiments in complex, nonlinear, contact-rich settings, such as the 37-dimensional, 8-DoF quadruped Ant environment, where even standard RL struggles without handcrafted reward shaping.

During the rebuttal, we extended our experiments to include additional ablation studies with SHAC/AHAC trained without our differentiable approach, as suggested by Reviewer QdnH, further emphasizing the consistent performance gains achieved by our method. We expanded our discussion to compare with constrained optimization techniques (e.g., barrier functions, MPC), signal temporal logic, and discrete techniques such as reward machines, and we justified our choice of LTL and differentiable reward machines. We also added discussions of the limitations of the differentiability assumption, including differentiable state labels, as well as potential extensions to discrete settings and stochastic transition functions. In addition, we elaborated on design trade-offs such as the choice of discount factors, signal functions, and activation functions, and we outlined several promising directions for future work, including hybrid partial differentiability and integrations with constrained optimization techniques.

We believe this paper lays the foundation for a new paradigm in learning from high-level linear temporal logic specifications in continuous domains, achieving both theoretical guarantees and empirical acceleration. By making automaton-based rewards differentiable, we enable seamless integration of symbolic task specifications into modern gradient-based RL pipelines. This capability, not addressed by existing methods, opens up new research directions at the intersection of formal methods, differentiable physics, and AI-driven long-horizon robotics, aligning closely with ICLR’s mission to advance the frontiers of machine learning.

---

### Meta-Review · Area_Chair_GUFN · 2026-01-05

**Summary:**

Please see the following part. Generally, I believe all concerns are largely addressed, and hence recommend for acceptance.

**Reviewer Concerns:**

The reviewers generally found the paper's idea of integrating Linear Temporal Logic (LTL) with differentiable simulation to be elegant and well-motivated. However, they raised several significant concerns regarding the scope, baselines, and theoretical assumptions:

(1). Experiments: Missing Ablations (from Reviewer QdnH). External Baselines (Reviewer kd6K)

(2).Theoretical Assumptions: Multiple reviewers (Ay51, QdnH) questioned if the framework could extend to discrete MDPs. The differentiability assumption is also questioned.  One reviewer noted that while the theory covers stochastic settings, the experiments were exclusively deterministic.

From my understandings, all these concerns are largely addressed. The only negative reviewer QdnH mainly questions about the differentiability assumption, which I believe is addressed.

**Reviewer Scores:**

n/a

---

### Decision · Program_Chairs · 2026-01-26

Accept (Poster)